# Diagnosing and Addressing Pitfalls in KG-RAG Datasets: Toward More Reliable Benchmarking

**Liangliang Zhang[1]  Zhuorui Jiang[2]  Hongliang Chi[1]  Haoyang Chen[1]  Mohammed Elkoumy[1]**
**Fali Wang[3]    Qiong Wu[4]    Zhengyi Zhou[4]    Shirui Pan[5]    Suhang Wang[3]    Yao Ma[1]**

[1]Rensselaer Polytechnic Institute    [2]University of Toronto    [3]Pennsylvania State University
[4]AT&T Chief Data Office    [5]Griffith University

{zhangl41,chih3,chenh29,elkoum,may13}@rpi.edu
zhuorui.jiang@mail.utoronto.ca, {qw6547,zz547k}@att.com
s.pan@griffith.edu.au, {fqw5095,szw494}@psu.edu

## Abstract

Knowledge Graph Question Answering (KGQA) systems rely on high-quality benchmarks to evaluate complex multi-hop reasoning. However, despite their widespread use, popular datasets such as WebQSP and CWQ suffer from critical quality issues, including inaccurate or incomplete ground-truth annotations, poorly constructed questions that are ambiguous, trivial, or unanswerable, and outdated or inconsistent knowledge. Through a manual audit of 16 popular KGQA datasets—including WebQSP and CWQ —we find that the average factual correctness rate is only 57%. To address these issues, we introduce KGQAGen, an LLM-in-the-loop framework that systematically resolves these pitfalls. KGQAGen combines structured knowledge grounding, LLM-guided generation, and symbolic verification to produce challenging and verifiable QA instances. Using KGQAGen, we construct KGQAGen-10k, a 10K-scale benchmark grounded in Wikidata, and evaluate a diverse set of KG-RAG models. Experimental results demonstrate that even state-of-the-art systems struggle on this benchmark, highlighting its ability to expose limitations of existing models. Our findings advocate for more rigorous benchmark construction and position KGQAGen as a scalable framework for advancing KGQA evaluation [1].

## 1 Introduction

Knowledge graph-based retrieval-augmented generation (KG-RAG) systems combine symbolic retrieval with generative reasoning, enabling question answering that requires both factual accuracy and structured inference [47, 12, 28, 21, 76, 56]. As these systems gain attention in both academic and industrial settings [15, 49, 77], benchmark datasets play a central role in measuring progress and guiding model development [73, 58, 79, 20, 10]. However, despite their central role in evaluation, little attention has been paid to the quality and reliability of these benchmarks themselves.

To better understand the limitations of existing benchmarks, we conduct a detailed manual inspection of 16 publicly available KGQA datasets including the widely adopted WebQSP and CWQ [73, 58, 5, 54, 39, 79, 4, 51, 62, 14, 23, 25, 20, 48, 8, 10]. A detailed summary of these datasets, along with our sampling and evaluation protocol, is provided in Appendix B. Our analysis reveals several recurring issues that compromise their utility for evaluating KG-RAG systems. These include

---

[1]KGQAGen:https://github.com/liangliang6v6/KGQAGen; KGQAGen-10k: https://huggingface.co/datasets/lianglz/KGQAGen-10k.

39th Conference on Neural Information Processing Systems (NeurIPS 2025) Track on Datasets and Benchmarks.

factually incorrect or outdated ground truth answers, and ambiguously phrased or trivially simple questions. In particular, WebQSP and CWQ, which serve as the dominant[2] evaluation benchmarks in recent KG-RAG research [72, 18, 71, 66, 78], exhibit serious deficiencies. Specifically, we found only 52% of sampled WebQSP examples and 49.3% of CWQ examples to be factually correct. Across all 16 datasets, the average correctness rate is just 57%, based on manual evaluation of over 1,000 question–answer pairs. A detailed summary of dataset quality is provided in Table 1. Additionally, many datasets rely on rigid exact-match metrics that penalize semantically correct answers expressed in alternative forms, further limiting their reliability [50, 45, 64].

Motivated by the issues, we propose KGQAGen, a framework for constructing high-quality benchmarks for KG-RAG systems. KGQAGen grounds question generation in a large, up-to-date Wikidata [63] as knowledge base and leverages a modular LLM-in-the-loop pipeline to ensure that each instance is factually correct and linguistically well-formed. Key components of the framework include iterative LLM-guided KG exploration and symbolic verification. Each question is grounded in a subgraph of the knowledge graph, which is iteratively expanded from a seed entity to include richer relational structures. This expansion enables the generation of more challenging and semantically complex questions. To ensure both efficiency and relevance, an LLM guides the expansion process by selecting informative entities and checking contextual sufficiency. Finally, symbolic verification using SPARQL ensures that the generated answers are correct and fully supported by the knowlege base. Together, these components enable the scalable generation of challenging, diverse, and reliable QA pairs.

We further use KGQAGen to generate a sample dataset KGQAGen-10k consisting of 10,787 QA pairs. KGQAGen-10k serves as a case study for investigating the quality, diversity, and characteristics of instances produced by the framework. Manual inspection of 300 samples reveals 96% factual accuracy. We analyze the dataset along several axes—including linguistic complexity, topic coverage that KGQAGen produces well-scoped, multi-hop questions with diverse structure and clear answer grounding. Moreover, we use KGQAGen-10k to benchmark a diverse set of models, including both pure LLMs and KG-RAG approaches. Our evaluation shows the difficulty of KGQAGen-10k: even SOTA models such as GPT-4.1 and recent KG-RAG systems such as GCR [34] and PoG [9] achieve only moderate performance. These results validate that the benchmark can effectively expose limitations in retrieval and reasoning, and demonstrate the utility of KGQAGen as a scalable, interpretable, and diagnostic tool for developing more capable KG-RAG systems. While KGQAGen-10k serves as a representative sample for our investigation, KGQAGen is fully modular and scalable, making it well-suited for constructing large-scale benchmarks with minimal human oversight.

**Contributions.** To summarize, our work makes three key contributions: (1) a systematic manual audit of 16 widely used KGQA datasets, uncovering critical quality and evaluation issues; (2) the development of KGQAGen, a scalable LLM-guided framework for generating challenging, grounded, and verifiable KGQA benchmarks; and (3) the construction of KGQAGen-10k, a 10K-scale dataset used to analyze question characteristics and benchmark a diverse set of KG-RAG models, revealing significant performance gaps and opportunities for future improvement.

## 2 Related Work

In this section, we briefly review prior work on KG-RAG and KGQA benchmark construction. We focus on the most relevant developments; a detailed discussion is provided in Appendix A.

**KG-RAG Methods.** KG-RAG systems enhance large language models by incorporating structured knowledge from KGs. Recent approaches such as RoG [33], GCR [34], and ToG [72] combine symbolic retrieval with multi-hop reasoning capabilities. Other frameworks, including DeCAF [74], DualR [29], and FRAG [18], focus on diverse strategies for integrating retrieval and generation. These efforts aim to improve factual accuracy and interpretability with external KG.

**KGQA Benchmarks.** Traditional KGQA datasets like WebQSP mainly rely on Freebase [7], which officially dumped since 2016. WebQuestions [6] and its successor WebQSP [73] generate dataset manually by collecting questions through Google Suggest API [38] and having Amazon MTurk workers annotate the ground truths. CWQ [58] extended this approach to handle more complex queries. To improve coverage and linguistic variety, subsequent benchmarks shifted to more diverse knowl-

---

[2]Around 30 papers on KG-RAG released between 2022 and 2025 adopted WebQSP and/or CWQ for evaluation, highlighting their widespread use. A detailed breakdown is provided in Appendix B

Table 1: **Systematic Audit of KGQA Datasets (Manual Verification Results).**

| Dataset | KG | Year | Sample / Total | Correctness (%) |
|---|---|---|---|---|
| WebQSP | Freebase | 2016 | 100 / 1639 | **52.00** |
| ComplexWebQuestions | Freebase | 2018 | 300 / 3531 | **49.33** |
| ComplexQuestions | Freebase | 2016 | 60 / 800 | **63.33** |
| GraphQuestions | Freebase | 2016 | 60 / 2607 | **70.00** |
| QALD | DBpedia | 2018 | 60 / 150 | **61.67** |
| MetaQA | WikiMovies | 2018 | 60 / 39093 | **20.00** |
| SimpleDBpediaQA | DBpedia | 2018 | 60 / 8595 | **43.33** |
| CSQA | Wikidata | 2018 | 60 / 27797 | **65.00** |
| LC-QuAD | DBpedia/Wikidata | 2017/2019 | 60 / 7046 | **38.34** |
| FreebaseQA | Freebase | 2019 | 60 / 3996 | **98.67** |
| CFQ | Freebase | 2020 | 60 / 239357 | **71.67** |
| GrailQA | Freebase | 2020 | 60 / 13231 | **30.00** |
| QALD-Plus | DB/Wikidata | 2022 | 60 / 136 | **63.33** |
| KQA-Pro | FB15k+Wikidata | 2022 | 60 / 11797 | **66.67** |
| DynamicKGQA | YAGO | 2025 | 60 / 40000 | **45.00** |

edge bases. `LC-QuAD 1.0&2.0` [62, 14] used a hybrid approach combining SPARQL templates and human annotation over DBpedia and Wikidata. `QALD-9` [39, 48] targeted multilingual QA evaluation, while `CSQA` [51] introduced conversational structures. `KQAPro` [8] emphasized compositional reasoning with program annotations. `MetaQA` [79] used a synthetic movie KG to assess multi-hop reasoning and robustness to paraphrasing. Other datasets addressed specific evaluation goals. `GrailQA` [20] focused on generalization—i.i.d., compositional, and zero-shot—using questions over Freebase. GeneralizableKGQA [24] extended this by re-splitting multiple datasets under shared evaluation settings. CBench [44] and SmartBench [42] analyzed datasets from the SPARQL structural complexity and linguistic diversity. Maestro [43] proposed a rule-based automatic construction framework, though its reliance on manually defined predicate rules limits its generality. More recently, scalable generation methods have emerged. CHATTY-Gen [40] introduced dialogue-style questions featuring coreference and ellipsis. `Dynamic-KGQA` [10] employed LLMs to adaptively generate QA instances from YAGO 4.5, but faced challenges from KG sparsity and hallucinated outputs. In summary, these existing efforts aim to enhance KGQA datasets from various perspectives—such as increasing linguistic diversity, enabling compositional reasoning, and improving scalability through automation. While valuable, few have systematically examined or addressed quality assurance. In contrast, our work focuses on factual correctness and verifiability.

# 3 Pitfalls of Existing KGQA Benchmarks

We identify two key issues with existing KGQA benchmarks for KG-RAG: (1) data quality problems, including inaccurate labels and trivial or ambiguous questions; and (2) rigid EM-based evaluation, which overlooks semantically correct but differently phrased answers.

## 3.1 Dataset Quality Issues

To assess the reliability of existing KGQA benchmarks, we manually inspected over 1,000 question-answer pairs sampled from 16 widely used datasets. A summary of findings is provided in Table 1. A description of the datasets and inspection protocol is included in appendix B. For each dataset, we randomly selected a representative subset of examples and evaluated them along three key dimensions: (1) factual correctness of the annotated answer, and (2) clarity and appropriateness of the question and (3) current evaludation of exact match shortcoming. We paid particular attention to the `WebQSP` and `CWQ` datasets, as they are the most dominant benchmarks in recent KGQA research. As shown in Table 1, for `WebQSP` and `CWQ`, we sampled 100 and 300 examples, respectively. For the remaining datasets, we uniformly sampled 60 examples. Overall, our inspection revealed substantial quality issues across many benchmarks. Notably, the most widely used `WebQSP` and `CWQ` datasets show serious quality issues. In `WebQSP`, only 52% of the sampled answers were judged correct, with quality issues including inaccurate answer and poor questions. Similarly, `CWQ`, which builds on `WebQSP` as its seed, achieved a correctness rate of just 49.33%, suffering from similar flaws along with additional issues due to increased question complexity. Beyond these two, we found that over half of the evaluated datasets had correctness rates below 60%, suggesting widespread challenges with annotation accuracy and question clarity. Next, we discuss the specific issues.

### 3.1.1 Inaccurate Ground Truth Answers.

A major issue across many KGQA datasets is the presence of inaccurate ground truth answers. Our detailed inspection reveals that such errors arise from distinct sources, which we categorize below.

- **Incorrect annotations** refer to cases where the labeled answer fails to align with the question's intent. For instance, in `WebQSP`, the question *"Where did Andy Murray start playing tennis?"* is incorrectly annotated with *2005*, a year rather than a location—demonstrating a mismatch between the expected answer type and the provided label.
- **Outdated answers** occur when the annotated response reflects facts that were once accurate but are no longer valid according to up-to-date knowledge. This is common for time-sensitive information, such as political positions, affiliations, or current locations. For instance, the question from `WebQSP` *"Who is the president of Peru now?"* lists *Ollanta Humala*, who was president from 2011-2016. This leads to penalizing correct model predictions that reflect up-to-date knowledge.
- **Incomplete annotations** happen when a question has multiple valid answers, but only one or a few are labeled as correct. This is common in open-ended or set-based questions. For instance, in `Dynamic-KGQA`, the question *"Which American actress is known for her notable work in a film where she also starred as an actor?"* is labeled with just *Lindsay Lohan*, even though others like Barbra Streisand and Angelina Jolie clearly qualify. As a result, models that return equally correct answers are unfairly penalized.

We provide additional examples of each type of issues in Appendix C.1.1. Although `FreeBaseQA` stands out with a high answer correctness rate (98.7%), a closer inspection reveals that many of its questions are overly simple and lack reasoning depth. We detail this issue in the next subsection.

### 3.1.2 Low-Quality or Ambiguous Questions

Another major limitation of existing KGQA datasets is the prevalence of low-quality or poorly constructed questions. In our analysis, these issues were observed in nearly all datasets to varying degrees. We categorize the common problems into three types.

- **Ambiguous phrasing** arises when questions lack sufficient context to uniquely identify a specific entity or relation. For example, the `WebQSP` question *"What does George Wilson do for a living?"* is inherently under-specified because it fails to distinguish which individual named George Wilson is being referred to. Multiple well-known figures share this name — including a fictional character from *The Great Gatsby*, a recurring comic strip character from *Dennis the Menace*, and several professional athletes, making it hard to answer accurately without additional contextual clues.
- **Low-complexity questions** require only shallow, one-hop retrieval, or string matching, offering little value in evaluating complex reasoning. This issue is especially common in `MetaQA` and `FreeBaseQA`. While `MetaQA` suffers from both low answer correctness (25%) and poor clarity, `FreeBaseQA` achieves a high correctness rate (98.7%). However, this high correctness appears to stem from the simplicity of the questions. To investigate this further, we used `GPT-4o` to answer directly the questions of `FreeBaseQA`, achieving 90. 39% accuracy in the evaluation of exact matches. The strong performance even without KG access suggests that the dataset contains mostly factoid, low-reasoning questions that are easily handled by powerful language models. These results reinforce the view that `FreeBaseQA`, despite its clean annotations, does not present a sufficient reasoning challenge for benchmarking KG-RAG systems.
- **Unanswerable, subjective, or ill-formed questions** are incompatible with structured KG-based reasoning. For example, `WebQSP` includes questions such as *"What to do today in Atlanta with kids?"* and *"What inspired Michael Jackson to become a singer?"*, both of which are inherently subjective and lack definitive, factual answers. Similarly, the `Dynamic-KGQA` question *"Which American university alumnus shares the same nationality as the creator of Kryptos?"* is poorly constructed, as any American university alumnus would automatically satisfy the condition.

We provide additional representative examples for each issue type in Appendix C.1.2.

## 3.2 Limitations of Exact-Match Evaluation

Existing KGQA benchmarks also suffer from shortcomings in their evaluation protocols. Most benchmarks rely on rigid exact-match criteria that fail to account for semantically correct answers expressed in different surface forms. This leads to false negatives when models generate correct answers that differ slightly from the annotated label. For example, in the `WebQSP` question *"What*

*is the Australian dollar called?"*, the ground truth answer is "AUD". A prediction of "Australian dollar", though semantically equivalent, would be considered incorrect under exact match. Similar mismatches arise from differences in formatting, paraphrasing, or entity naming conventions (e.g., "Germany" vs. "Federal Republic of Germany"). Additional examples are provided in Appendix C.2.

## 4  `KGQAGen`: A Framework for Grounded KGQA Dataset Construction

As discussed in the previous section, existing KGQA benchmarks suffer from quality and reliability issues that limit their utility for evaluating KG-RAG systems. To address this, we introduce `KGQAGen`, a modular framework for constructing high-quality QA datasets that are both semantically grounded and verifiable. `KGQAGen` grounds each question in explicit KG evidence and leverages LLMs to assist with subgraph expansion, question generation, and answer validation—enabling the scalable creation of challenging and reliable benchmark instances.

The overall process of `KGQAGen` is illustrated in Figure 1. The framework consists of three main stages: (1) **Seed Subgraph Initialization**: The process begins by selecting a seed entity and constructing a local subgraph by retrieving related facts from the KG. This subgraph provides the initial context for reasoning. (2) **Question Generation through Iterative LLM-Guided Subgraph Expansion**: To support non-trivial, multi-hop questions, we iteratively expand the subgraph by traversing neighboring entities and relations. This involves alternating between KG traversal and LLM evaluation. At each step, the subgraph is expanded by exploring neighboring entities and relations within the KG. After each expansion, an LLM is prompted to evaluate whether the subgraph contains sufficient information to support a well-formed, multi-hop question. If not, the expansion continues. Once the subgraph is judged sufficient, the LLM generates a natural language question, identifies the corresponding answer set, extracts a minimal *supporting subgraph*, and constructs the associated SPARQL query. (3) **Answer Validation and Refinement**: The final stage ensures that the generated answer set is correct and fully grounded in the KG. To achieve this, the generated SPARQL query is executed against the knowledge base to retrieve the actual answers. If the results match the generated answer set, the instance is accepted. An ablation study presented in Appendix E further confirms the importance of each component in `KGQAGen`. Both LLM-guided subgraph expansion and SPARQL-based verification are shown to be essential for producing accurate and well-grounded QA pairs. Before we proceed to detail the framework, we introduce the notation used throughout this section.

**Notations.** We denote a KG as a set of triples $\mathcal{G} \subseteq \mathcal{E} \times \mathcal{R} \times \mathcal{E}$, where $\mathcal{E}$ is the set of entities and $\mathcal{R}$ is the set of relations. Each triple $\langle s, p, o \rangle \in \mathcal{G}$ represents a factual statement with subject $s \in \mathcal{E}$, predicate $p \in \mathcal{R}$, and object $o \in \mathcal{E}$. For a given seed entity $e \in \mathcal{E}$, we denote the associated subgraph as $\mathcal{G}_e \subseteq \mathcal{G}$, and its state after $t$ rounds of expansion as $\mathcal{G}_e^{(t)}$.

### 4.1  Seed Subgraph Initialization.

To ensure topic diversity and coverage across different domains, the selection of seed entities plays a critical role. We draw seed entities from the Wikipedia Vital Articles [3], which includes a curated set of globally relevant topics spanning a broad range of domains. This provides a principled starting point for constructing diverse and non-trivial question-answer pairs.

For each selected seed entity $e$, we construct an initial local subgraph $\mathcal{G}_e^{(0)}$ by sampling a fixed number (e.g. 15) of its 1-hop neighbors. The resulting subgraph includes the seed and its sampled neighbors, along with the triples connecting them. For example, in Figure 1, starting from the seed entity *Nobel Prize*, we include sampled 1-hop neighbors such as *Nobel Peace Prize*, *Norway*, *Alfred Nobel*, and a few other related entities, which together form the context for further expansion and question generation. This initialization step helps constrain the knowledge scope while maintaining enough structure to support meaningful reasoning.

### 4.2  Question Generation through Iterative LLM-Guided Subgraph Expansion

To generate high-quality questions that require structured, multi-hop reasoning, it is important to go beyond shallow, single-fact queries. This requires subgraphs that are both semantically rich and

---

[3] `https://en.wikipedia.org/wiki/Wikipedia:Vital_articles` (accessed April 2, 2025)

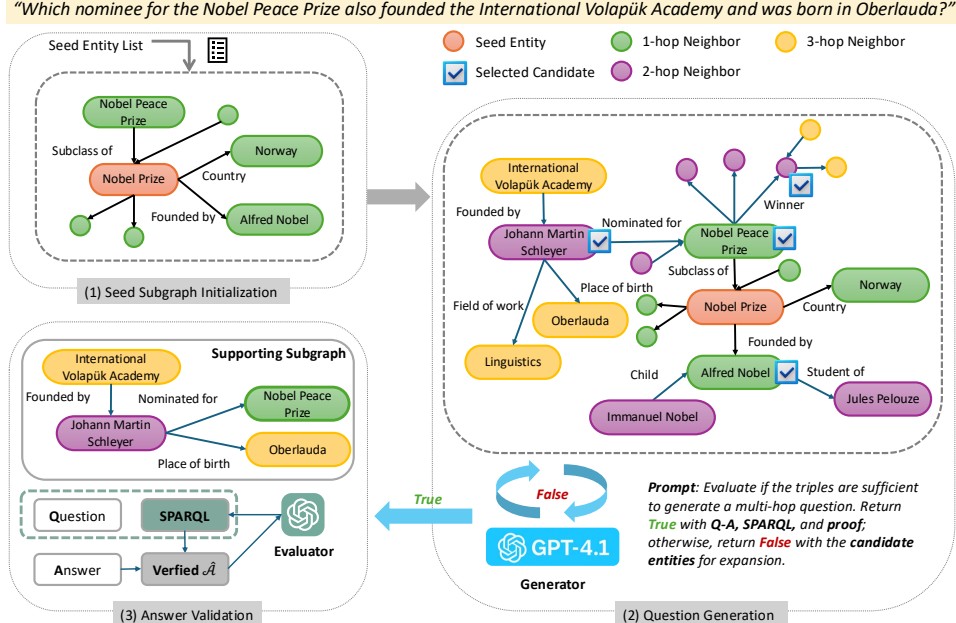

Figure 1: KGQAGen framework.

structurally diverse. Therefore, we aim to expand the initialized *seed subgraph* $\mathcal{G}_e^{(0)}$ to include more informative paths and relation patterns that support deeper reasoning over the KG. A straightforward way is to expand the seed subgraph $\mathcal{G}_e^{(0)}$ using multi-hop traversal methods like BFS or DFS. However, in densely connected KGs, such unrestricted expansion quickly becomes impractical: even a few hops from a high-degree entity can yield thousands of nodes and edges, leading to bloated subgraphs that are too large to process efficiently. Moreover, these fully expanded subgraphs often contain irrelevant or weakly connected facts, making it difficult to maintain question quality.

Partial traversal offers a more practical alternative, typically limiting expansion to 1–2 hops or selectively sampling deep paths. But without intelligent guidance, selected paths may be arbitrary or loosely related, weakening semantic coherence. As a result, such subgraphs tend to be noisy, hard to interpret, and ill-suited for generating meaningful multi-hop questions.

To generate questions that are both challenging and well-formed, it is necessary to strike a balance between *coverage* (including diverse, relevant entities) and *depth* (capturing reasoning chains of sufficient complexity). To this end, we adopt an iterative expansion strategy guided by an LLM. Rather than relying on fixed-depth traversal, we allow the LLM to evaluate whether the current subgraph contains enough information to support a valid question and to suggest targeted expansion directions when additional context is needed. Once the subgraph is considered sufficient, the LLM proceeds to generate a QA instance. In the following subsections, we describe the two core components.

### 4.2.1 LLM-Guided Iterative Subgraph Expansion

Given an initialized subgraph $\mathcal{G}_e^{(0)}$ centered on a seed entity $e$, our goal is to iteratively expand it to accumulate sufficient contextual knowledge for question generation. In this subsection, we describe the $(t+1)$-th iteration of this process, assuming that the subgraph $\mathcal{G}_e^{(t)}$ generated in the previous iteration is considered insufficient by the LLM. In this case, in the previous iteration $t$, the LLM also produces an *Exploration Set* $\mathcal{C}_e^{(t)}$, consisting of entities within $\mathcal{G}_e^{(t)}$ that are identified as requiring further exploration. Next, we first describe the one-hop expansion procedure for iteration $t + 1$. The LLM-based sufficiency judgment and the generation of the *Exploration Set* are discussed in detail in the Subsection of **Sufficiency Check and Exploration Set Generation.**.

**One-Hop Expansion in Iteration** $t + 1$. Given the subgraph $\mathcal{G}_e^{(t)}$ and the *Exploration Set* $\mathcal{C}_e^{(t)}$ identified by the LLM in iteration $t$, we expand the subgraph by incorporating additional knowledge from the global KG $\mathcal{G}$. Specifically, we perform a one-hop expansion around each entity $c \in \mathcal{C}_e^{(t)}$ to gather new facts that may help support the generation of more complex and informative questions.

For each entity $c$ in the *Exploration Set* $\mathcal{C}_e^{(t)}$, we randomly sample a small number of its 1-hop neighbors (e.g., 10–15) and include the corresponding triples that connect these neighbors to $c$. This process adds a bounded amount of new information to the subgraph while keeping the expansion efficient and focused. The updated subgraph is defined as: $\mathcal{G}_e^{(t+1)} = \mathcal{G}_e^{(t)} \cup \mathrm{SampledTriples}(\mathcal{C}_e^{(t)})$, where $\mathrm{SampledTriples}(\mathcal{C}_e^{(t)})$ denotes the union of the selected 1-hop triples associated with the entities in $\mathcal{C}_e^{(t)}$. This one-hop expansion ensures that the subgraph grows in a controlled manner. As a result, the resulting subgraph $\mathcal{G}_e^{(t+1)}$ captures richer relational context while preserving locality and relevance to the question generation task.

**Sufficiency Check and Exploration Set Generation.** In this subsection, we check whether the updated subgraph $\mathcal{G}_e^{(t+1)}$ includes enough information to support a meaningful and well-scoped question. To do this, we prompt an LLM to perform two tasks: (1) evaluate whether the current subgraph is sufficient for question generation, and (2) if not, suggest a set of entities for further exploration. We refer to this set as the *Exploration Set* $\mathcal{C}_e^{(t+1)}$.

For *Sufficiency Checking*, we require that the subgraph support at least 2-hop reasoning and contain semantically meaningful paths sufficient to generate a well-scoped, unambiguous question. It should go beyond shallow or generic facts and provide just enough context to support a non-trivial, grounded question. The *Exploration Set* guides further expansion toward subgraph regions more likely to yield such questions. We prioritize semantically specific and structurally central entities—such as notable people, events, or awards—over broad or generic ones like countries. For example, in Figure 1, we prefer expanding on *Nobel Peace Prize* or *Alfred Nobel* over *Norway*.

To support both sufficiency checking and exploration set generation, we prompt `GPT-4.1` with a comprehensive template. The prompt contains clear instructions and concrete examples illustrating sufficient vs. insufficient subgraphs, as well as good and bad exploration targets. The detailed prompt can be found in Appendix D.1. We will revisit this prompt in Section 4.2.2, where we describe how it also handles question and answer generation.

### 4.2.2 Question Generation from the Finalized Subgraph

Once a subgraph is judged to be sufficient, we denote it as $\mathcal{G}_e^*$, representing the finalized subgraph used for question generation. Given the finalized subgraph $\mathcal{G}_e^*$, we prompt the LLM to generate a complete question-answer instance. The outputs include: (1) a natural language question $q_e$, (2) an answer set $\mathcal{A}_e$, (3) a *supporting subgraph* $\mathcal{P}_e \subseteq \mathcal{G}_e^*$, and (4) a corresponding SPARQL query $\mathcal{Q}_e$. These components are generated jointly to maintain consistency and alignment with the reasoning required by the question. The *supporting subgraph* captures the minimal set of facts needed to justify the answer. It also facilitates error diagnosis within `KGQAGen`. The SPARQL query is expected to verify the answer set with the KG.

To ensure the generated question is meaningful and challenging, we impose several constraints. The question must be answerable using only the given finalized subgraph $\mathcal{G}_e^*$ and involve at least two-hop reasoning. It should be specific, unambiguous, and self-contained. The question should be phrased naturally and fluently. These requirements are enforced in the unified prompt (detailed in Appendix D.1) introduced in the previous Section, which also handles sufficiency checking and exploration set selection. These three tasks are tightly linked: sufficiency checking determines whether to proceed with question generation or continue expanding the subgraph. By handling them together, we ensure that each decision is made in a shared context, leading to more coherent and aligned outputs.

### 4.3 Answer Validation and Refinement

The goal of this stage is to ensure that each generated question–answer pair is faithfully grounded in the KG. To achieve this, we use the SPARQL query as a formal verification tool: it must successfully retrieve the intended answer when executed over the KG.

Given a generated instance consisting of a question $q_e$, answer set $\mathcal{A}_e$, supporting subgraph $\mathcal{P}_e$, and SPARQL query $\mathcal{Q}_e$, we first execute $\mathcal{Q}_e$ over the KG to retrieve a result set $\hat{\mathcal{A}}_e$. We then compare this result set to the LLM-generated answer set $\mathcal{A}_e$. If $\mathcal{A}_e = \hat{\mathcal{A}}_e$, we accept the instance as valid. If the two answer sets do not match, we prompt a lightweight LLM (`GPT-4o-mini`) to revise the SPARQL query (see Appendix D.2 for the prompts). We then re-execute the revised query and repeat

the validation. This revision loop continues for up to three attempts. If the revised SPARQL query returns results that match the original answer set, we retain the instance and include the revised query in the final output. Otherwise, the instance is discarded. This conservative filtering ensures that only verifiable and KG-grounded instances are retained. Additionally, since each question is paired with an executable query, the dataset can be periodically revalidated as the KG evolves—making `KGQAGen`-generated benchmarks both accurate at creation and maintainable over time.

## 5 `KGQAGen-10k`: A Sample Dataset Generated by `KGQAGen`

To demonstrate the practical capabilities of `KGQAGen`, we construct `KGQAGen-10k`, a 10K-scale KGQA dataset with `KGQAGen`. This dataset serves as a representative output that illustrates the effectiveness of `KGQAGen` in producing challenging, well-grounded, and verifiable question–answer instances. Note that the design of `KGQAGen` is highly modular and scalable. With minimal human effort, it can be readily extended to generate substantially larger datasets across diverse knowledge domains. In this section, we focus on the construction and analysis of `KGQAGen-10k`.

**Dataset Construction.** To construct `KGQAGen-10k`, we begin by selecting 16,000 seed entities from Wikipedia's Level-5 Vital Articles, which provide broad topical coverage and global relevance. All questions are grounded in the most up-to-date dump of Wikidata[4], a large and publicly accessible knowledge base with comprehensive entity and relation coverage. Using `KGQAGen`, we generate 15,451 question–answer pairs. After applying answer validation to remove examples with non-executable or inconsistent SPARQL queries, we obtain 10,787 verified instances. We refer to this dataset as `KGQAGen-10k`.

**Manual Quality Assessment.** To assess the factual accuracy of `KGQAGen-10k`, we conducted a manual audit of 300 randomly sampled question–answer pairs. Following the same annotation protocol used in Section 3, each example was carefully verified. We found that 289 out of 300 examples (96.3% ) are correct. This result suggests that the QA instances generated by `KGQAGen` are highly reliable and well-grounded in verifiable knowledge. We defer analysis of question difficulty to Section 6, where we benchmark several LLM and KG-RAG models. A case study of the generated questions is provided in Appendix F.2.

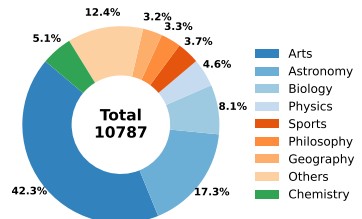

Figure 2: Topic Distribution

**Statistics and Properties.** We analyze `KGQAGen-10k` along two key dimensions: linguistic complexity and topic coverage: (1) *Linguistic Complexity:* Most questions (61.1%) are 16–30 words long, showing moderate complexity. A third exceed 30 words, indicating deeper reasoning. Only 7.5% are short factoid queries, highlighting the dataset's focus on rich, natural questions. For answer set size, 84.5% of the questions yield a single answer, while 9.7% return three or more answers. (2) *Structural Difficulty:* Following DyVal [81], we compute six graph-based metrics for each supporting subgraph—number of nodes, edges, average degree, depth (hops), width (maximum frontier), and extra links (non-chain shortcuts). Across all 10,787 examples, 98% require 2–5 hops, 84% contain 5–30 entities, 83% have 4–28 relations, 79% exhibit width 4–20, and 92% are fully connected. These statistics confirm that `KGQAGen-10k` consists of diverse, non-trivial reasoning graphs with substantial structural complexity. A full summary table is provided in Appendix F.1. (3) *Topic Coverage:* The dataset also provides broad semantic coverage. Figure 2 shows that `KGQAGen-10k` covers a broad range of topics. The most represented categories are Arts (42.3%) and Astronomy (17.3%), followed by STEM fields (16% combined), Sports, Geography, and Philosophy.

## 6 `KGQAGen-10k` as a Benchmark for KG-RAG Models

In this section, we use `KGQAGen-10k` to benchmark a variety of KG-RAG models and LLMs.

**Experiment Setup.** We benchmark all models on `KGQAGen-10k` using a standardized split of 8,629 training, 1,079 development, and 1,079 test examples. Most prior work evaluates KGQA systems using an exact string match between predicted and reference answers. However, as highlighted in

---

[4]`https://dumps.wikimedia.org/wikidatawiki/` (accessed April 2, 2025)

Table 2: Performance on `KGQAGen-10k`. EM = Exact Match; LASM = LLM-Assisted Semantic Match; LLM-SP = LLM with Supporting Subgraph as input.

| Type | Model | Accuracy | | Hit@1 | | F1 | | Precision | | Recall | |
|------|-------|----------|------|-------|------|------|------|-----------|------|--------|------|
| | | EM | LASM | EM | LASM | EM | LASM | EM | LASM | EM | LASM |
| Pure LLM | `LLaMA-3.1-8B-Instruct` | 8.87 | 11.91 | 9.27 | 12.42 | 8.97 | 11.98 | 9.27 | 12.42 | 8.87 | 11.81 |
| | `LLaMA2-7B` | 6.88 | 12.32 | 7.23 | 12.88 | 6.95 | 12.34 | 7.23 | 13.72 | 6.88 | 11.96 |
| | `Mistral-7B-Instruct-v0.2` | 24.98 | 32.34 | 26.51 | 34.38 | 25.36 | 33.20 | 26.60 | 34.85 | 24.98 | 32.72 |
| | `GPT-4o-mini` | 32.34 | 42.49 | 34.11 | 44.39 | 32.74 | 42.91 | 34.11 | 44.86 | 32.34 | 42.35 |
| | `GPT-4` | 42.38 | 51.37 | 44.95 | 54.49 | 43.01 | 52.32 | 45.13 | 55.33 | 42.38 | 51.48 |
| | `DeepSeek-Chat` | 42.48 | 51.84 | 45.51 | 55.24 | 43.17 | 52.64 | 45.60 | 55.79 | 42.48 | 51.78 |
| | `GPT-4o` | 45.29 | 54.21 | 47.91 | 57.46 | 45.89 | 54.93 | 48.01 | 57.83 | 45.29 | 54.11 |
| | `GPT-4.1` | **47.43** | **56.96** | **50.05** | **59.96** | **48.03** | **57.72** | **50.05** | **60.33** | **47.43** | **56.95** |
| RAG-based | `RoG (LLaMA2-7B)` | 20.10 | 27.28 | 21.32 | 28.92 | 17.75 | 24.26 | 17.65 | 24.79 | 20.10 | 27.16 |
| | `GCR (LLaMA-3.1 + GPT-4o)` | 49.37 | 58.96 | 52.46 | 62.84 | 49.30 | 58.88 | 50.61 | 60.76 | 49.37 | 59.18 |
| | `ToG (GPT-4o)` | 49.65 | 59.89 | 52.55 | 63.02 | 50.38 | 60.73 | 53.01 | 64.23 | 49.65 | 59.76 |
| | `PoG (GPT-4o)` | **50.67** | **60.18** | **54.03** | **63.95** | **51.47** | **61.30** | **54.31** | **64.78** | **50.67** | **60.34** |
| LLM-SP | `LLaMA2-7B (w/ SP)` | 69.79 | 73.79 | 73.12 | 77.76 | 70.43 | 74.55 | 72.81 | 77.67 | 69.79 | 73.76 |
| | `GPT-4o (w/ SP)` | **82.46** | **84.89** | **89.62** | **92.22** | **84.07** | **86.75** | **89.81** | **93.23** | **82.46** | **84.95** |

Section 3, this metric often fails to capture semantically valid predictions that differ in surface form. To address this limitation, we introduce **LLM-Assisted Semantic Match (LASM)**, a lightweight verification mechanism that activates only when a prediction fails the exact match. LASM uses `GPT-4o-mini` to assess semantic equivalence between predictions and gold answers (details in Appendix G). We report Accuracy, Hit@1, Precision, Recall, and F1 under both Exact Match (EM) and LASM for direct comparison.

**Evaluated Models.** We evaluate three categories of models on `KGQAGen-10k`: (1) *Pure LLMs:* This group includes a range of open-source and commercial models—`LLaMA-3.1-8B-Instruct`, `LLaMA2-7B`, `Mistral-7B-Instruct-v0.2`, `GPT-4o-mini`, `GPT-4`, `GPT-4o`, `GPT-4.1`, and `Deepseek-Chat`. These models receive only the natural language question without any retrieval or external grounding; (2) *KG-RAG Models:* We include recent KG-RAG models such as `RoG`, `GCR`, `ToG`, and `PoG`. These models use the KG as an external retrieval source, either by incorporating retrieved triples, augmenting the prompt with symbolic paths, or integrating topology-aware context; and (3) *LLM with Supporting Subgraph (LLM-SP).* To assess performance under perfect retrieval, we include `LLaMA2-7B` and `GPT-4o` models that are directly given the associated *supporting subgraph* of the questions from the dataset construction process. These setups simulate perfect retrieval and help isolate the reasoning capabilities from retrieval effectiveness. See Appendix G for more details of baseline models.

**Results and Analysis.** Table 2 summarizes the performance of all evaluated models across standard metrics under both EM and LASM protocols. We make the following key observations: (1) Even capable LLMs such as `GPT-4.1` and recent KG-RAG models like `GCR` and `PoG` achieve only moderate performance on `KGQAGen-10k`, showing the non-trivial nature of the questions and the challenges they pose for current methods. (2) LASM consistently yields higher reported performance across all models over EM, indicating that many predictions marked incorrect under exact match are actually semantically correct. This highlights the importance of incorporating semantic-aware evaluation to more accurately assess QA performance. (3) Model capability strongly correlates with performance: larger and more recent LLMs such as `GPT-4.1` substantially outperform smaller models like `LLaMA2-7B`, showing the importance of both knowledge coverage and reasoning ability. (4) KG-RAG models achieve noticeable gains over their corresponding LLM backbones. For example, `RoG` improves upon `LLaMA2-7B` by over 10 points, while `GCR`, `ToG`, and `PoG` outperform `GPT-4o` by approximately 4 points. These results confirm that incorporating external KG-derived context enhances QA performance. However, the overall improvement remains moderate, suggesting that retrieval components in current KG-RAG systems are still suboptimal and warrant further enhancement. (5) LLM-SP models—LLMs provided with the ground truth *supporting subgraph*—achieve the strongest performance by a substantial margin. For instance, `LLaMA2-7B (w/ SP)` reaches LASM accuracy 73.8%, outperforming not only its base model `LLaMA2-7B` but all other larger LLMs. Similarly, `GPT-4o (w/ SP)` sees its LASM accuracy rise from 54.2% to 84.9% compared with `GPT-4o`. These results highlight the key role of high-quality retrieval in KG-RAG systems and suggest that retrieval remains a major bottleneck limiting current KG-RAG systems.

**Cross-Dataset Comparison.** To further contextualize the results, we evaluate representative KG-RAG models on two established KGQA benchmarks, WebQSP [73] and CWQ [58], in addition to

our `KGQAGen-10k`. While models such as `RoG`, `GCR`, `ToG`, and `PoG` achieve high Hit@1 scores on We-bQSP and CWQ (e.g., 85.7 and 92.2, respectively), their performance drops sharply on `KGQAGen-10k` (21.3–54.0), reflecting its higher reasoning complexity and stricter grounding requirements. Detailed results are provided in Appendix G.3. This consistent decline across models highlights that legacy benchmarks often overestimate true reasoning ability, whereas `KGQAGen-10k` offers a more challenging and diagnostic testbed for evaluating KG-RAG systems.

## 7 Conclusion

In this paper, we identified major quality issues in existing KGQA benchmarks through a detailed manual audit, including inaccurate answers and poorly constructed questions. To address this, we introduced `KGQAGen`, a framework for building well-grounded, verifiable QA datasets at scale. Using this framework, we created `KGQAGen-10k`, a 10K-example benchmark that challenges both general-purpose LLMs and KG-RAG models. Our results show that even strong models struggle on `KGQAGen-10k`, highlighting the need for better retrieval and reasoning mechanisms. They also demonstrate the value of `KGQAGen` as a scalable and effective approach for constructing challenging, high-quality benchmarks to support future progress for KG-RAG systems.

**Limitations:** While `KGQAGen` enables scalable and verifiable KGQA dataset construction, it relies on the availability and accuracy of the underlying KG (e.g., Wikidata). Errors or gaps in the KG can limit the quality of generated questions. In addition, our pipeline depends on LLM capabilities and prompt design; although we mitigate hallucination risks via symbolic verification, the quality of subgraph selection and question phrasing may still be influenced by LLM variability.

## Acknowledgments

This research is supported by the National Science Foundation (NSF) under grant numbers NSF-2406647 and NSF-2406648. It is also supported by the National Artificial Intelligence Research Resource (NAIRR) Pilot and the Delta advanced computing and data resource, which is supported by the National Science Foundation under award NSF-OAC-2005572.

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

# NeurIPS Paper Checklist


# Appendices

## A  Related Works

This section provides comprehensive technical details and broader context for the related work summarized in Section 2, focusing on the methodological foundations and evaluation challenges that motivate our framework.

### A.1  KG-RAG Methods.

Knowledge graph-based retrieval-augmented generation systems address hallucination and factual grounding limitations in large language models by integrating structured symbolic knowledge into the generation process. These systems retrieve relevant subgraphs based on input queries to serve as structured context, effectively combining broad parametric knowledge with the precision and verifiability of knowledge bases.

Recent developments have produced several distinct architectural approaches addressing different aspects of the integration challenge. Graph-guided reasoning methods, exemplified by Reasoning-on-Graph (RoG) [33], enable interpretable multi-step reasoning by having language models explicitly

verbalize their traversal through knowledge graph structures, generating relation paths that are then grounded in actual graph connections. This approach provides both answer generation and explanation capabilities, making the reasoning process more transparent and debuggable. Extensions include GNN-RAG [36], which incorporates graph neural networks to better capture structural patterns in retrieved subgraphs. In contrast, constrained generation approaches like Graph-Constrained Reasoning (GCR) [34] focus on ensuring faithful outputs by incorporating explicit graph-based constraints into the decoding process, using KG-Trie structures to restrict generation to only those paths that exist in the knowledge base.

Modular and memory-augmented architectures represent another important direction. Frameworks like FRAG [18] propose adaptive combinations of different retrieval and generation components, while Generate-on-Graph [72] treats the language model as both an agent and a knowledge graph component, enabling interactive knowledge graph expansion during question answering. Memory-augmented approaches such as MemQ [71] introduce dedicated memory modules that separate language model reasoning from knowledge graph tool usage. More sophisticated integration strategies include DeCAF [74], which jointly generates natural language answers and corresponding logical forms, and ReknoS [66], which introduces abstract relationship reasoning through super-relations for complex compositional queries.

## A.2  KGQA Benchmarks.

Despite major progress in KGQA dataset construction, existing benchmarks exhibit persistent limitations that hinder effective evaluation of modern KG-augmented retrieval systems. Early human-curated datasets such as WebQuestions [6] and ComplexQuestions [5] focused on capturing authentic user queries and introducing compositional constraints, but these resources are dominated by simple factoid questions or contain ambiguities and incomplete annotations, making them insufficient for testing models that require deeper reasoning and precise answer grounding.

To address these shortcomings, semi-automated and automated approaches have become prevalent. Hybrid pipelines like LC-QuAD [62, 14] and GraphQuestions [54] use SPARQL templates and graph-structured logic to guide question generation, with subsequent human editing for naturalness. While this increases structural diversity and complexity, the template-based nature often leads to unnatural or overly constrained question styles. More recent fully automated frameworks, including Maestro [43], CHATTY-Gen [40], and DYNAMIC-KGQA [10], attempt to scale question generation via rules, popularity heuristics, or dialogue simulation, yet these methods still struggle to balance natural language fluency, logical completeness, and answer correctness for challenging multi-hop or compositional queries.

The evolution of KGQA benchmarks reflects the field's growing complexity, progressing from simple factoid questions to sophisticated multi-hop reasoning scenarios. Early benchmarks established foundational paradigms: WebQuestions pioneered collecting natural language questions through search engine suggestions with human annotation, WebQuestionsSP [73] added SPARQL annotations aligned with Freebase, and ComplexWebQuestions [58] advanced complexity by programmatically generating SPARQL queries with logical constructs. However, these early datasets relied heavily on Freebase, which ceased maintenance in 2016, motivating migration efforts to more sustainable knowledge bases like DBpedia and Wikidata, though such migrations often introduced conceptual mismatches and alignment difficulties. Specialized evaluation objectives drove targeted benchmark development. GrailQA [20] introduced systematic evaluation of generalization scenarios including i.i.d., compositional, and zero-shot settings, while KQAPro [8] emphasized compositional reasoning through explicit program annotations and MetaQA [79] focused on multi-hop reasoning using controlled movie domain knowledge graphs. Conversational benchmarks like CSQA [51] introduced multi-turn dialogue structures, and recent work incorporated dialogue-style questions with coreference and ellipsis to better reflect real-world query patterns. Meanwhile, evaluation methodology improvements from CBench [44] and SmartBench [42] provided comprehensive analysis of SPARQL query complexity and linguistic diversity.

Despite these varied approaches, our systematic audit reveals that annotation quality remains a persistent challenge across benchmarks, with even widely-used datasets like WebQSP and CWQ exhibiting correctness rates below 60 percent. Furthermore, the predominant reliance on exact-match evaluation metrics fails to capture semantic equivalence, leading to systematic underestimation of model

capabilities and underscoring the need for more rigorous benchmark construction methodologies that prioritize both annotation accuracy and semantically-aware evaluation protocols.

# B KGQA Datasets and Inspection Protocol

## B.1 Basic Dataset Statistic

This section provides additional context for the dataset analysis presented in Section 3, reviewing existing KGQA benchmarks and their construction methodologies. We categorize these datasets based on their generation approaches and underlying knowledge bases. These datasets differ in construction methods (manual vs. automated), target knowledge graphs (Freebase [7], DBpedia [3], Wikidata [63], YAGO [55]), and reasoning complexity. Some emphasize compositional or multi-hop reasoning, while others focus on multilingual support, dialogue capabilities, or logical form mapping. Our review examines these datasets to understand their construction approaches, key features, and relevance for evaluating knowledge graph-enhanced question answering systems.

**GraphQuestions** [54] is constructed through a semi-automated pipeline that begins with the generation of structured queries over Freebase. These queries are designed to capture varied reasoning functions such as counting, superlatives, and conjunctions, as well as different structural complexities and answer cardinalities. Queries that are infrequent or low-quality are filtered using web statistics. The remaining set is then verbalized into natural language using pre-defined templates, creating a dataset well-suited for evaluating semantic parsing and logical compositionality.

**WebQuestionsSP** [73] builds upon the WebQuestions [6] dataset by providing SPARQL annotations aligned with Freebase. It employs a modular annotation interface that guides workers to select topic entities, predicates, and relevant constraints, ensuring consistent semantic representation. The dataset offers executable queries, enabling precise evaluation of models' ability to map natural language questions to structured representations.

**ComplexWebQuestions** [58] extends WebQuestionsSP by programmatically generating more complex SPARQL queries that incorporate logical constructs such as conjunctions, comparatives, and superlatives. These queries are automatically translated into machine-generated questions, which are then paraphrased into natural language by crowdworkers. The resulting dataset challenges QA models with deeper compositional reasoning and varied surface forms.

**QALD-9** [39] is a multilingual benchmark constructed through manual curation, where annotators write natural language questions and align them with SPARQL queries over DBpedia. It emphasizes diversity in question types and supports multiple languages. Its extension, QALD-9-Plus, translates these questions into additional languages and maps them to Wikidata, enabling cross-lingual and cross-knowledge-base evaluation.

**MetaQA** [79] frames question answering as a latent variable problem, where both the topic entity and the reasoning path are unobserved. Built over a movie-related knowledge graph, the dataset includes 1-hop, 2-hop, and 3-hop questions, designed to evaluate multi-step reasoning. A variational neural framework is used to learn both entity disambiguation and graph-based reasoning simultaneously.

**SimpleDBpediaQA** [4] remaps the widely used SimpleQuestions dataset from Freebase to DBpedia. This conversion involves aligning entities and predicates via owl:sameAs links and rewriting SPARQL queries to account for conceptual mismatches such as directionality, ambiguity, and redirections. The resulting dataset enables QA over an actively maintained knowledge base while preserving the simplicity of the original benchmark.

**CSQA** [51] introduces a large-scale, multi-turn dialogue dataset for complex question answering over Wikidata. It combines crowd-sourced and in-house annotations to generate over 200K question-answer pairs, including clarification, comparison, and logical reasoning within dialogue context. CSQA is designed to benchmark conversational agents that require memory and contextual understanding.

**LC-QuAD 1.0 and 2.0** [62, 14] are created through a three-stage semi-automated pipeline. First, SPARQL queries are instantiated using templates and selected entities. Second, these queries are mapped to question templates. Finally, crowdworkers paraphrase them into fluent natural language. The datasets support a wide range of question types, including boolean, temporal, compositional, and multi-relation queries, and target DBpedia and Wikidata respectively.

**FreeBaseQA** [23] compiles trivia-style factoid question–answer pairs from public sources and aligns them with Freebase triples using a two-way entity linking approach. Human annotators then validate relevance and correctness. The dataset features over 54K entity-answer pairs covering 28K unique natural questions, offering high linguistic diversity and a challenging alternative to SimpleQuestions.

**Compositional Freebase Questions** [25] is developed to assess compositional generalization. It begins by generating logical forms and corresponding SPARQL queries using a unified grammar. Natural questions are written to express these logical forms. The dataset is then split using a divergence-based strategy to maximize the gap in compositional structure between training and test sets, enabling rigorous evaluation of generalization.

**GrailQA** [20] constructs question–answer pairs through a structured pipeline involving logical form generation over Freebase, expert-written canonical questions, and crowd-sourced paraphrases. It includes three evaluation splits: i.i.d., compositional, and zero-shot, making it suitable for analyzing generalization across different reasoning complexities and linguistic expressions.

**QALD-9 Plus** [48] enhances QALD-9 by adding more questions, refining SPARQL annotations, and expanding multilingual support. The updated version improves coverage of complex queries and diverse linguistic phenomena, making it more suitable for modern multilingual QA systems.

**KQA Pro** [8] is constructed by generating compositional reasoning programs (KoPL) and corresponding SPARQL queries over a curated knowledge base. These are paraphrased into natural questions via crowdsourcing. The dataset supports fine-grained evaluation across logical reasoning categories such as multi-hop inference, comparison, boolean logic, and temporal constraints.

**Dynamic-KGQA** [10] departs from static benchmarks by generating question–answer pairs on-the-fly. It samples compact, semantically coherent subgraphs from YAGO 4.5 [55] and uses LLMs to produce multi-hop questions grounded in these subgraphs. This design minimizes data leakage and supports controlled, reproducible QA benchmarking with adaptive complexity.

## B.2 Recent KG-RAG Publications with WebQSP and/or CWQ

WebQSP [73] and CWQ [58] have emerged as the primary benchmarks for empirical evaluation in LLM-based knowledge graph question answering. Table 3 presents a chronological summary of recent LLM-based KGQA models, indicating for each whether WebQSP and CWQ were used for evaluation and providing a brief description of the model's main approach. Notably, nearly 30 KG-RAG models released between 2022 and 2025 have adopted one or both datasets as central components of their experimental protocols, despite the quality issues identified in Section 3.

Early LLM–KG hybrids, including ReaRev [35] and DECAF [74], set a precedent by reporting results on both datasets, but typically focused on Hit@1 as the main metric. As the field progressed, it became clear that Hit@1 alone could obscure over-generation and other qualitative issues. This recognition prompted a shift in the community: subsequent agent-style models such as ToG [57], RoG [33], ChatKBQA [32], and KD-CoT [65] began to report full precision, recall, and F1 scores on both benchmarks, enabling more nuanced and meaningful comparison. By 2024 and 2025, evaluation on WebQSP and CWQ had become an established standard for the field. State-of-the-art systems—such as GCR [34], Effi-QA [13], GNN-RAG [36], PoG [9], CLEAR-KGQA [67], and ReKnoS [66]—universally benchmarked on these datasets before extending to additional corpora or domain-specific tasks. Even in research targeting specialised knowledge graphs, models like RARoK [75] and Efficient-G-Retriever [53] included WebQSP or CWQ for calibration and comparability. As captured in Table 3, the adoption trajectory of these datasets not only reflects their ubiquity but also highlights their role in shaping rigorous and transparent evaluation practices for the next generation of KGQA systems.

## B.3 Manual Inspection Protocol

We evaluate a broad selection of prominent KGQA benchmarks, including , and others commonly used in recent KGQA literature. Each dataset provides a set of natural language questions paired with ground-truth answers, and, where available, supporting triples or subgraphs from the underlying knowledge graph (e.g., Freebase, Wikidata, DBpedia, WikiMovies and some subsets).

Table 3: Chronological summary of recent LLM-based KGQA models, with dataset usage based on reported experiments (✓).

| Author [Citation] | Model | WebQSP | CWQ | Year | Short Introduction |
|---|---|---|---|---|---|
| Mavromatis et al. [35] | ReaRev | ✓ | ✓ | 2022 | LLM + GNNs refine reasoning on incomplete graphs. |
| Yu et al. [74] | DECAF | ✓ | ✓ | 2022 | Joint answer/logical form decoding from free-text retrieval. |
| Sun et al. [57] | ToG | ✓ | ✓ | 2023 | LLM agent explores KGs via beam search for deep, interpretable reasoning. |
| Luo et al. [33] | RoG | ✓ | ✓ | 2023 | Relation-grounded KG paths guide LLM reasoning with explanations. |
| Luo et al. [32] | ChatKBQA | ✓ | ✓ | 2023 | LLM-generated logical forms, improved with KG retrieval. |
| Wang et al. [65] | KD-CoT | ✓ | ✓ | 2023 | External KG knowledge injected into CoT reasoning. |
| Liu et al. [30] | DualR | ✓ | ✓ | 2024 | GNN for structural reasoning, frozen LLM for semantic reasoning. |
| Luo et al. [34] | GCR | ✓ | ✓ | 2024 | KG-Trie constrains LLM decoding for logic-faithful KG reasoning. |
| Dong et al. [13] | Effi-QA | ✓ | ✓ | 2024 | Iterative LLM planning, KG exploration, and self-reflection for QA. |
| Mavromatis et al. [36] | GNN-RAG | ✓ | ✓ | 2024 | GNN-based subgraph reasoning with LLM in RAG pipeline. |
| Xu et al. [70] | READS | ✓ | ✓ | 2024 | LLM decomposes KGQA into retrieval, pruning, inference. |
| Li et al. [26] | DoG | ✓ | ✓ | 2024 | LLM generates "well-formed chains" via constrained decoding. |
| Fang et al. [17] | KARPA | ✓ | ✓ | 2024 | LLM pre-plans, matches KG paths, reasons in training-free manner. |
| Xu et al. [72] | GoG | ✓ | ✓ | 2024 | LLM agent selects, generates, reasons on incomplete KGs. |
| Zhan et al. [75] | RARoK | ✓ | ✓ | 2024 | RAG-augmented CoT for complex medical KGQA. |
| Li et al. [27] | SubgraphRAG | ✓ | ✓ | 2024 | MLP + triple-scoring for efficient subgraph extraction. |
| Fang et al. [16] | DARA | ✓ | | 2024 | LLM decomposes and grounds formal KG queries. |
| Hu et al. [22] | GRAG | ✓ | | 2024 | Text-to-graph, retrieves/prunes subgraphs for RAG. |
| Xiong et al. [69] | Interactive-KBQA | ✓ | ✓ | 2024 | LLM agent generates SPARQL via multi-turn KB interaction. |
| Dehghan et al. [12] | EWEK-QA | ✓ | ✓ | 2024 | Web retrieval + KG triple extraction for citation-based QA. |
| Chen et al. [9] | PoG | ✓ | ✓ | 2024 | Self-correcting LLM planner for decomposed KGQA. |
| Wen et al. [67] | CLEAR-KGQA | ✓ | ✓ | 2025 | Interactive clarification and Bayesian inference for ambiguity. |
| Tan et al. [59] | Path-Over-Graphs | ✓ | ✓ | 2025 | LLM agent explores/prunes multi-hop KG paths. |
| Wang et al. [66] | ReKnoS | ✓ | ✓ | 2025 | Aggregates "super-relations" for LLM forward/backward reasoning. |
| Xu et al. [71] | MemQ | ✓ | ✓ | 2025 | Memory module separates LLM reasoning from KG tool use. |
| Gao et al. [18] | FRAG | ✓ | ✓ | 2025 | Modular KG-RAG adapts retrieval to query complexity. |
| Shen et al. [52] | RwT | ✓ | ✓ | 2025 | LLM-guided MCTS refines KG reasoning chains. |
| Solanki et al. [53] | Efficient-G-Retriever | ✓ | | 2025 | Attention-based subgraph retriever for LLM-aligned RAG. |
| Tang et al. [60] | GGI-MAB | ✓ | ✓ | 2025 | Multi-armed bandit adapts RAG retrieval for KGQA. |
| Zhang et al. [78] | TrustUGA | ✓ | | 2025 | Unified Condition Graph, two-level LLM querying. |

For our systematic audit, we followed a unified sampling and review protocol. For WebQSP and CWQ—the most widely adopted benchmarks—we randomly sampled 100 and 300 test examples, respectively, to ensure sufficient coverage of prevalent error types. For all other datasets, we sampled 60 examples per set to enable a balanced cross-dataset comparison. Each sampled item was independently reviewed by two annotators with KGQA expertise, resolving disagreements through discussion.

During inspection, we assessed each example along three main dimensions: (1) factual correctness of the annotated answer; (2) clarity and appropriateness of the question; and (3) faithfulness of the supporting SPARQL, where available. We flagged instances with incorrect, incomplete, or ambiguous annotations, as well as questions that were underspecified, trivial, or unanswerable.

# C    Pitfalls of Existing KGQA Benchmarks

Many benchmarks are anchored to deprecated resources like Freebase, and even migration efforts to Wikidata [46] introduce further inconsistencies in entity mappings and answer verification. Most critically, almost all existing datasets are evaluated using rigid exact match (EM) metrics, which fail to recognize semantically equivalent answers phrased differently. In summary, current KGQA datasets face two central challenges: (1) data quality issues, including inaccurate, incomplete, or artificial annotations, and (2) narrow evaluation protocols that do not capture true semantic correctness. These limitations underscore the need for new benchmarks—such as KGQAGen—that emphasize both annotation quality and robust, semantically-aware evaluation.

## C.1    Current KGQA Dataset Issues

This appendix presents a detailed case study of data quality issues involved the most widely used KGQA benchmarks, drawing from our broader review of 16 major datasets. For each dataset, we randomly sampled question-answer pairs and performed careful manual verification following the evaluation criteria in Section B. By presenting both problematic examples and the reasoning behind their classification, this analysis provides concrete, case-based evidence that complements the aggregate statistics reported in Table 1 and discussed in Section 3.

Our review identifies three principal categories of data quality problems, as defined in Section 3.1: **Inaccurate Ground Truth Answers**, **Low-Quality or Ambiguous Questions**, and **Limitations of Exact-Match Evaluation**. These recurring issues—rooted in annotation, question design, and evaluation protocols—can seriously undermine the reliability of KGQA benchmarks. A comprehensive breakdown, along with additional annotated examples for each dataset, is provided in the supplementary materials at the shared repository[5].

### C.1.1    Inaccurate Ground Truth Answers

Following the categorization presented in Section 3, we provide additional examples for each type of answer annotation error identified in our analysis: A major challenge in KGQA benchmarks is the prevalence of ground truth annotation errors, including incorrect, incomplete, or outdated answers. These undermine evaluation reliability and can mislead model training.

**Incorrect annotations.**    Incorrect annotations occur when the labeled answer does not actually address the question or contains factual mistakes.

- **ID: WebQTest-273**
  **Question:** When did Michael Jordan return to the NBA?
  **Answer:** 1984
  **Issue:** 1984 is the year of his NBA debut, not his return. The correct answer should be 1995.

- **ID: QALD9Plus-120**
  **Question:** Who is the daughter of Bill Clinton married to?
  **Answer:** Chelsea Clinton
  **Issue:** Answer is Chelsea Clinton herself, not her spouse. The correct answer should be Marc Mezvinsky.

- **ID: GrailQA-2101221015000**
  **Question:** The 1912–13 Scottish Cup season is part of what sports league championship?
  **Answer:** 1913 Scottish Cup Final
  **Issue:** The answer refers to a final match, not the correct league entity ("Scottish Cup").

- **ID: DynamicKGQA-26720**
  **Question:** Which actor starred in both 'The Hospital' and was born in the same country as Albert Einstein?
  **Answer:** George C. Scott
  **Issue:** George C. Scott was born in the United States, while Einstein was born in Germany. The answer does not satisfy the nationality constraint.

---

[5]https://drive.google.com/drive/folders/1hH-NxqbUkOSeLC3q1ifLpq6iOlByait4?usp=drive_link

**Outdated answers.** Outdated answers reflect facts that were once correct but have become obsolete due to real-world changes and key issue is the outdated knowledge source.

- **ID: WebQTest-182**
  **Question:** Who is Khloe Kardashian's husband?
  **Answer:** Lamar Odom
  **Issue:** The answer is outdated; Khloe Kardashian and Lamar Odom divorced in 2016.

- **ID: CWQ-1182_ac67410188d0f2258139a3c84773885e**
  **Question:** Who is the current queen of the location whose time zone is Central Western Time Zone?
  **Answer:** Elizabeth II
  **Issue:** The answer is outdated. The Central Western Time Zone (UTC+8:45) corresponds to a small region in Australia. Since Queen Elizabeth II passed away in 2022, the current monarch of Australia is King Charles III.

- **ID: KQAPro-14**
  **Question:** What city's population is 6,690,432?
  **Answer:** Dalian
  **Issue:** The question lacks a specified time period, making the answer potentially outdated. Additionally, there is no country constraint, which may lead to ambiguity.

- **ID: FreebaseQA-eval-836**
  **Question:** Who is currently the creative director at the house of Chanel?
  **Answer:** Karl Lagerfeld
  **Issue:** Karl Lagerfeld passed away in 2019 and no longer holds this position.

- **ID: ComplexQuestions-33**
  **Question:** Who is Greece's leader now?
  **Answer:** Karolos Papoulias
  **Issue:** The answer is outdated; Karolos Papoulias served as president until 2015. The correct answer should be the current leader.

- **ID: MetaQA-46**
  **Question:** What were the release dates of films directed by the director of [Rosemary's Baby]?
  **Answer:** 1986, 1948, 1992, 1982, 1994, 1979, 1999, 1965, 1974, 1967, 1971, 2002, 1988, 2005, 1976, 2011, 2010, 2013
  **Issue:** The answer is incomplete and outdated; it does not include the director's most recent films, such as a 2023 release.

**Incomplete annotations.** Incomplete annotations occur when the gold set omits other valid answers, penalizing models that provide equally correct alternatives.

- **ID: CWQ-124_7360e892294860c6ef7ad9a10e540e1b**
  **Question:** What movie did the author who published editions for *Notes from My Travels* direct?
  **Answer:** Unbroken
  **Issue:** The annotation is incomplete. The book *Notes from My Travels* was written by Angelina Jolie, who has directed multiple films, including *In the Land of Blood and Honey* (2011), *Unbroken* (2014), *By the Sea* (2015), and *First They Killed My Father* (2017).

- **ID: GrailQA-2100689004000**
  **Question:** What fossil specimen dates from the Eocene?
  **Answer:** Darwinius masillae
  **Issue:** The annotation is incomplete; many Eocene fossils (e.g., Moeritherium lyonsi) would also be valid answers.

- **ID: DynamicKGQA-14927**
  **Question:** Which American professor at the University of Wisconsin–Madison worked in the same city where Charles V. Bardeen died?
  **Answer:** E. Ray Stevens
  **Issue:** The annotation is incomplete; any American professor at UW–Madison would satisfy the condition, not just E. Ray Stevens.

- **ID: DynamicKGQA-24330**
  **Question:** Which athlete from Novosibirsk shares their nationality with the owner of the United Shipbuilding Corporation?
  **Answer:** Yevgeni Nikolayevich Andreyev
  **Issue:** The annotation is incomplete; multiple Russian athletes from Novosibirsk could be correct.

- **ID: GraphQuestions-281000202**
  **Question:** The British coat of arms has which heraldic supporters included?
  **Answer:** Lion
  **Issue:** The annotation is incomplete; both a lion (left) and a unicorn (right) are supporters. Only listing "lion" is insufficient.

- **ID: KQAPro-72547**
  **Question:** What person has a notable work titled "The Scorpion King," which was produced by Vince McMahon?
  **Answer:** Dwayne Johnson
  **Issue:** Incomplete annotation; any individual involved in the production could be considered correct, not just Dwayne Johnson.

### C.1.2 Low-Quality or Ambiguous Questions

Low-quality questions include those that are ambiguous, overly simple, or fundamentally unanswerable. We highlight key cases by dataset.

**Ambiguous phrasing.**    Ambiguous questions make it unclear what the intended answer should be, undermining evaluation objectivity.

- **ID: GrailQAPlus-2101990009000**
  **Question:** Which automotive designer designed NA?
  **Answer:** Koichi Hayashi, Tom Matano, Bob Hall
  **Issue:** The meaning of "NA" is ambiguous and could refer to multiple models or entities.

- **ID: GraphQuestions-358000401**
  **Question:** sci came after what video game engine?
  **Answer:** Adventure Game Interpreter
  **Issue:** The question is ambiguous as "sci" does not have a defined meaning or Wikidata label.

- **ID: GrailQA-3200295001000**
  **Question:** What is the number of theater plays that are in mystery?
  **Answer:** 1
  **Issue:** The question is too vague and does not provide enough context to determine the correct answer.

- **ID: WebQTest-108**
  **Question:** What was the book written by Charles Darwin?
  **Answer (length: 153, unique: 94):** On evolution, The Autobiography of Charles Darwin, The Voyage of the Beagle, The Origin of Species, ...
  **Issue:** The question is ambiguous—Darwin wrote many books, but the prompt refers to "the" book. The ground truth annotation is also excessively broad and noisy, containing 153 answers (including duplicates), which undermines reliable evaluation.

**Low-complexity questions.**    Low-complexity questions can be solved by simple lookup or direct retrieval, offering little test of reasoning or multi-hop capabilities.

- **ID: QALD9Plus-143**
  **Question:** What is the area code of Berlin?
  **Answer:** 030
  **Issue:** This is a straightforward 1-hop fact lookup with no reasoning required.

- **ID: FreebaseQA-eval-1536**
  **Question:** Who wrote the 1970 book 'Future Shock'?

**Answer:** Alvin Toffler
**Issue:** This is a basic factoid query that tests only surface-level knowledge.

- **ID: FreebaseQA-eval-2363**
  **Question:** In which ocean is the island of Madeira?
  **Answer:** Atlantic Ocean
  **Issue:** This is a simple fact lookup with no reasoning challenge.

- **ID: CSQA-7**
  **Question:** Which sex does Wolfgang Brandstetter belong to?
  **Answer:** male
  **Issue:** This is a 1-hop lookup question with no requirement for reasoning ability.

- **ID: SimpleDBpedia-17597**
  **Question:** What was Karl Dönitz's place of death?
  **Answer:** Schleswig Holstein
  **Issue:** This is a simple factual retrieval, requiring no reasoning.

**Unanswerable, subjective, or ill-formed questions.** Such questions may rest on false premises, omit crucial context, or rely on subjective interpretations.

- **ID: GrailQA-2104467004000**
  **Question:** The time zone from UTC of 12.75 has been offset what number of times?
  **Answer:** 1
  **Issue:** The question is uninterpretable; UTC+12.75 is not a standard offset, and the phrasing lacks clarity.

- **ID: QALD9Plus-81**
  **Question:** Butch Otter is the governor of which U.S. state?
  **Answer:** [Missing Ground Truth]
  **Issue:** Unanswerable in the present; Butch Otter no longer holds that position.

- **ID: FreebaseQA-eval-3290**
  **Question:** What is measured in Scoville units?
  **Answer:** Pungency
  **Issue:** Subjective; the question could accept "spiciness" or "pungency," but only one is annotated as correct.

- **ID: GraphQuestions-462000201**
  **Question:** Find the bearers of the coat of arms granted by queen.
  **Answer:** Western Australia
  **Issue:** The question does not specify which queen or which coat of arms, making it ambiguous and unanswerable.

- **ID: KQAPro-3**
  **Question:** Which has lower elevation above sea level, Bristol or Jerusalem whose ISNI is 0000 0001 2158 6491?
  **Answer:** Bristol
  **Issue:** Problematic: the Jerusalem referenced is a musician, not a location. Multiple cities named Bristol exist, with no way to determine which is intended.

## C.2   Limitations of Exact-Match Evaluation

Existing KGQA benchmarks are further limited by their reliance on rigid exact-match evaluation protocols. Such criteria do not accommodate semantically correct answers that are phrased differently from the annotated ground truth. As a result, models are often penalized for generating correct answers that differ only in surface form, leading to false negatives and an underestimation of true model performance.

- **ID: QALD9-174**
  **Question:** Who is the novelist of the work a song of ice and fire?
  **Ground Truth:** George_R._R._Martin
  **Issue:** Other semantically correct forms such as "George Raymond Richard Martin," "George R. R. Martin" (with or without punctuation), "G. R. R. Martin," "George RR

Martin," "Martin, George R. R.," "George R. Martin," or "G.R.R. Martin" are equally valid. Exact-match evaluation penalizes correct answers that differ in surface form or formatting.

- **ID: QALD9-114**
  **Question:** How big is the earth's diameter?
  **Ground Truth:** 1.2742e+07
  **Issue:** Acceptable answers include "12,742 km," "12,742,000 meters," "about 7,918 miles," "$1.2742 \times 10^7$ meters," and "approximately 12,700 kilometers." Variations in units, notation, or approximation are all reasonable, but exact match evaluation may reject them as incorrect.

- **ID: CSQA-60**
  **Question:** Which nucleic acid sequence encodes Ufm1-specific protease 2?
  **Ground Truth:** Ufsp2
  **Issue:** Other valid forms include "Ufsp2 gene," "the gene encoding Ufm1-specific protease 2," "gene symbol: UFSP2," or Ensembl/NCBI identifiers. Exact match allows only the annotated form, penalizing equally correct alternatives.

- **ID: WebQTest-6**
  **Question:** Where is JaMarcus Russell from?
  **Ground Truth:** Mobile
  **Issue:** Answers such as "Mobile, Alabama," "the city of Mobile," "Mobile (city)," "Mobile, AL," or "JaMarcus Russell was born in Mobile, Alabama" all convey the same information, but may not be accepted unless they match the ground truth exactly.

- **ID: FreebaseQA-eval-607**
  **Question:** Who wrote the 1990 Booker Prize winner Possession?
  **Ground Truth:** a. s. byatt
  **Issue:** Other correct answers like "Antonia Susan Byatt," "Dame A. S. Byatt," or "Antonia Byatt" are semantically equivalent, but only the annotated form is accepted under exact match.

## D    Prompt Templates and Generation Examples

This section provides detailed documentation of the prompt templates used in KGQAGen, along with representative examples of generated questions and error cases. We first present the core generator prompt that guides question construction and knowledge sufficiency checking. We then describe the simplified evaluator prompt used during answer validation. Finally, we analyze example outputs and common error patterns to illustrate the framework's capabilities and limitations.

### D.1    Question Generation Prompt

The generator component of KGQAGen utilizes a carefully designed prompt template to ensure high-quality question generation. The prompt consists of input specifications and structured generation rules that guide the LLM in producing well-formed question-answer instances.

The input format specifies RDF triples from Wikidata, where each triple contains an entity label with its Q-ID, a predicate label with its P-ID, and another entity label with Q-ID. The LLM evaluates whether the given subgraph provides sufficient information for generating a meaningful KGQA instance. The generation rules enforce five key requirements for question construction. (1) Reasoning complexity demands that generated questions involve at least 2-hop logical reasoning paths. The framework prioritizes specific, instance-level entities while avoiding generic categories. Factual constraints are incorporated only when necessary for disambiguation or meaningful answer space reduction. (2) Entity selection criteria require that candidate entities must be concrete instances rather than abstract types. The system favors entities that enable meaningful multi-hop paths through affiliations, awards, locations, or temporal relationships, while avoiding generic class-level concepts. (3) Question difficulty ensures that questions are designed to require structured knowledge graph reasoning by incorporating inverse relations, numerical constraints, comparative logic, or set-based conditions. The framework employs factual filters to reduce ambiguity and ensures questions cannot be answered through general knowledge alone. (4) Natural language quality mandates that questions must use natural, fluent phrasing typical of real user queries. The prompt enforces self-contained and concise formulation while avoiding references to underlying data structures or unnecessary repetition. (5) Semantic clarity requires that all generated questions must unambiguously specify their intended

answers. The prompt explicitly prohibits vague or underspecified formulations that could lead to multiple valid interpretations.

The LLM returns a JSON response indicating either insufficiency with candidate entities for expansion or a complete question-answer instance containing the natural language question, answer set, supporting proof triples, and corresponding logical constraints. The full prompt template is provided below:

---

**Prompt**

```
You are given a small set of RDF triples from Wikidata.
Format:  Each triple is a 3-item array:  ["<label> (<Q-ID>)", "<predicate>
(<P-ID>)", "<label> (<Q-ID>)"]
Triples:  {triples}
Your task is to determine whether this subgraph is sufficient to support a
challenging and non-trivial question for a knowledge graph question answering
(KGQA) benchmark.
Guidelines:
1.  Reasoning Depth
        • Prefer questions requiring at least 2-hop reasoning.
        • Avoid generic topics or subclass chains-focus on instance-level,
          specific entities.
        • Use factual constraints (e.g., date, affiliation) only when needed to
          disambiguate the answer or add meaningful specificity.
        • Do not over-constrain-include only what is necessary to yield a specific
          answer.
2.  Entity Selection and Expansion
        • Focus on concrete, instance-level entities (e.g., Q7186), not types like
          Q5 (human) or Q11424 (film).
        • Avoid generic classes like ''scientist'', ''award'', or ''event'', and
          relations like ''subclass of'' or ''instance of''.
        • Prefer entities and paths supporting deeper reasoning-e.g., affiliations,
          recognitions, or spatiotemporal links.
3.  Difficulty
        • Encourage inverse relations, comparative logic, date/number filters, or
          set membership.
        • Ensure the answer cannot be derived from general knowledge alone.
        • The subgraph must contain all supporting information to answer the
          question.
4.  Naturalness
        • Phrase the question as a fluent, self-contained query a user might ask.
        • Avoid references to the input format (e.g., "triples", "given data").
        • Do not use phrases like:
            – "from the given data"
            – "among these entities"
            – "listed here"
5.  Clarity
        • The question must be unambiguous and logically imply a unique, specific
          answer.
        • Avoid vague or underspecified language.
Output Format:
If the graph is not sufficient, return:  { "sufficient":  false, "candidate":
[<QID>, ..., <QID>] }
If sufficient, return:  { "sufficient":  true, "question":  "<natural-language
question>", "answer":  ["<answer-label (QID)>", "..."], "proof":  [ ["<label
(QID)>", "<predicate (PID)>", "<label (QID)>"], ...  ]  }
Return strict JSON only - no commentary.
```

---

**Few-shot Example**

```
Example 1: Triples: [ ["Johann Martin Schleyer (Q12712)", "nominated for
(P1411)", "Nobel Peace Prize (Q35637)"], ["International VolapŎ0fck Academy
(Q3358168)", "founded by (P112)", "Johann Martin Schleyer (Q12712)"], ["Johann
Martin Schleyer (Q12712)", "place of birth (P19)", "Oberlauda (Q885402)"] ]
Output: { "sufficient": true, "question": "Who among the nominees for the
Nobel Peace Prize was also the founder of International VolapŎ0fck Academy?",
"answer": ["Johann Martin Schleyer (Q12712)"], "proof": [ ["Johann Martin
Schleyer (Q12712)", "nominated for (P1411)", "Nobel Peace Prize (Q35637)"],
["International VolapŎ0fck Academy (Q3358168)", "founded by (P112)", "Johann
Martin Schleyer (Q12712)"] ] }

Example 2: Triples: [ ["Karakalpakstan (Q484245)", "capital (P36)",
"Nukus (Q489898)"], ["Karakalpakstan (Q484245)", "shares border with (P47)",
"Mangystau Region (Q238931)"], ["Karakalpakstan (Q484245)", "official language
(P37)", "Karakalpak (Q33541)"], ["Karakalpakstan (Q484245)", "country (P17)",
"Uzbekistan (Q265)"] ]
Output: { "sufficient": false, "candidate": ["Q33541", "Q489898", "Q238931"]
}

Example 3: Triples: [ ["Astronomy and Astrophysics (Q752075)", "publisher
(P123)", "EDP Sciences (Q114404)"], ["Astronomy and Astrophysics (Q752075)",
"editor (P98)", "Thierry Forveille (Q46260676)"], ["Zeitschrift für Astrophysik
(Q3575110)", "followed by (P156)", "Astronomy and Astrophysics (Q752075)"] ]
Output: { "sufficient": true, "question": "What astronomical journal,
published by EDP Sciences and edited by Thierry Forveille, succeeded Zeitschrift
für Astrophysik as its immediate follower?", "answer": ["Astronomy and
Astrophysics (Q752075)"], "proof": [ ["Astronomy and Astrophysics (Q752075)",
"publisher (P123)", "EDP Sciences (Q114404)"], ["Astronomy and Astrophysics
(Q752075)", "editor (P98)", "Thierry Forveille (Q46260676)"], ["Zeitschrift
für Astrophysik (Q3575110)", "followed by (P156)", "Astronomy and Astrophysics
(Q752075)"] ] }
```

This structured prompt design ensures consistent generation of high-quality, diverse, and well-specified question-answer pairs for the benchmark dataset. The prompt template balances multiple objectives: maintaining reasoning complexity, ensuring natural language quality, and guaranteeing answer specificity. By enforcing these requirements through explicit rules and format specifications, we enable systematic generation of challenging yet well-formed KGQA instances.

## D.2 SPARQL Validation Prompt

The validation component uses a focused prompt template for verifying and refining SPARQL queries. This streamlined prompt specifically addresses query correction when execution results differ from intended answers, ensuring both syntactic correctness and semantic alignment.

The validation process operates on the principle of iterative refinement. When a generated SPARQL query fails to return expected results or returns empty result sets, the validation component engages a lightweight language model to diagnose and correct the query. This approach recognizes that initial query generation may suffer from syntactic errors, incorrect entity identifiers, or inappropriate query structure that prevents successful execution against the Wikidata endpoint.

The prompt template emphasizes simplicity and executability in query revision. Rather than attempting complex query transformations, the validation component focuses on ensuring that revised queries use only essential triple patterns and avoid unnecessary complexity that might introduce additional failure points. The template specifically discourages the use of optional clauses and filter conditions unless they are strictly necessary for answering the question, as these constructs often lead to query execution failures or unexpected empty results. Additionally, the validation process enforces structural requirements that ensure compatibility with the Wikidata query service. All revised queries must terminate with a single SERVICE wikibase:label clause to retrieve human-readable English labels for entities, maintaining consistency with the expected output format. The prompt also mandates syntactic validity and direct executability at the official Wikidata SPARQL endpoint, ensuring that corrected queries can be verified immediately. The output format maintains strict JSON formatting

requirements to facilitate automated processing. The validation component returns only the corrected SPARQL query without additional commentary or explanation, enabling seamless integration into the broader validation pipeline. This focused approach allows for rapid iteration and correction when initial query generation produces non-executable or semantically misaligned queries.

Through this validation mechanism, KGQAGen ensures that all retained question-answer pairs are grounded in verifiable SPARQL queries that can be executed against the knowledge base. This constraint provides a strong guarantee of answer correctness and enables ongoing validation as the underlying knowledge graph evolves over time.

```
Prompt

You are given a SPARQL query over Wikidata that returned no results.
Question: {question}
Original SPARQL: {sparql}
Your task is to revise the query so that it returns valid results from Wikidata.
Revision Guidelines:
• Use only essential triple patterns.  Avoid OPTIONAL and FILTER clauses unless
  strictly necessary.
• The query must end with a single SERVICE wikibase:label clause to retrieve
  English labels.
• Ensure the query is syntactically valid and directly executable at https:
  //query.wikidata.org.
Output Format:  Return a single JSON object in the exact format below - no
commentary, no markdown:

{
  "correct_sparql": "<REVISED SPARQL QUERY HERE>"
}
```

This structured prompt design ensures consistent generation of high-quality, diverse, and well-specified question-answer pairs for the benchmark dataset. The prompt template balances multiple objectives by maintaining reasoning complexity, ensuring natural language quality, and guaranteeing answer specificity. By enforcing these requirements through explicit rules and format specifications, we enable systematic generation of challenging yet well-formed KGQA instances.

# E   Ablation Study on `KGQAGen` Design

**Setup.** We perform an ablation study to quantify the contribution of core components in the `KGQAGen-10k` generation pipeline. In our framework, questions are generated through iterative LLM-guided subgraph expansion combined with symbolic SPARQL-based verification. To isolate the effect of each component, we consider three configurations: (A1) random subgraph selection replacing LLM-guided expansion, (A2) alternative generation LLMs, and (A3) inclusion or exclusion of SPARQL-based answer validation. A3 is treated as a cross-setting axis applied to all variants. We randomly sample 100 seed entities and report two metrics: *generation success rate* (percentage of seeds producing valid questions) and *end-to-end success rate* (percentage producing fully validated QA pairs).

| Setting | Generation Rate (%) | End-to-End Rate (%) | Validation Effect (%) |
|---|---|---|---|
| Full KGQAGen (GPT-4.1) | 94 | 77 | +17 |
| A1: Random Subgraph | 100 | 62 | +15 |
| A2-1: LLaMA-3-70B | 57 | 35 | +4 |
| A2-2: GPT-4o-mini | 65 | 41 | +6 |

Table 4: Ablation results on 100 randomly selected seed entities. "Validation Effect" denotes the improvement in valid QA pairs after applying SPARQL verification.

**Findings.** The full KGQAGen pipeline achieves a 94% generation success rate and a 77% end-to-end validation rate. Removing LLM-guided subgraph selection (A1) maintains full generation coverage but lowers validation to 62%, as random sampling often introduces extraneous entities that cause

ambiguity. To assess the effect of model choice, we replace GPT-4.1 with smaller LLMs under the same setup. LLaMA-3-70B generates 57 questions with 35 validated (61% of generated), while GPT-4o-mini produces 65 questions with 41 validated (63% of generated). Smaller models frequently simplify multi-hop reasoning to single-hop relations or omit relational constraints. In contrast, the full KGQAGen pipeline with GPT-4.1 yields 82 generated and 77 validated questions, achieving both higher coverage and correctness. These results demonstrate that model capacity and structure-aware subgraph guidance jointly determine data quality, while SPARQL-based symbolic verification mitigates hallucinations and constrains potential biases.

To better understand the failure modes of symbolic verification, we analyzed 420 instances filtered out by the SPARQL-based validation stage. The main error types include: empty query results (166 cases, 39.5%), excessively large answer sets (184 cases, 43.8%), and answer mismatches where the annotated answer was absent from the SPARQL results (70 cases, 16.7%). These issues primarily stem from incomplete or ambiguous question–answer pairs, limited KG coverage, or challenges in SPARQL query formulation. Future work will strengthen the validation module with tighter constraints, clarification prompts, and finer-grained error classification to improve explainability and systematically address validation failures.

# F   `KGQAGen-10k` **Analysis**

## F.1   `KGQAGen-10k` **Statistics**

**Structural Complexity Metrics.** Following DyVal [81], we measure six structural aspects of the supporting subgraphs for all 10,787 examples in `KGQAGen-10k`: (1) number of nodes, (2) number of edges, (3) average degree, (4) reasoning depth (hops), (5) width (maximum frontier size), and (6) extra links (non-chain connections). These metrics collectively capture both the scale and topological difficulty of multi-hop reasoning paths.

| Measure | Distribution Range | Percentage of Examples |
|---|---|---|
| Depth (hops) | 2–5 | 98% |
| Nodes (entities) | 5–30 | 84% |
| Edges (relations) | 4–28 | 83% |
| Width (max frontier) | 4–20 | 79% |
| Reachability | Fully connected | 92% |

Table 5: Graph-based reasoning complexity analysis for `KGQAGen-10k` following DyVal-style metrics. The dataset exhibits diverse and non-trivial structural properties, demonstrating its suitability for evaluating reasoning over complex knowledge graphs.

Across these dimensions, most questions involve 2–5-hop reasoning chains over compact yet densely connected subgraphs. The high connectivity (92%) and wide range of entity and relation counts illustrate that `KGQAGen-10k` contains structurally rich and diverse reasoning contexts, providing a challenging evaluation setting for KG-RAG and KGQA models.

## F.2   **A Case Study of** `KGQAGen-10k`

An audit of 300 randomly sampled question–answer pairs from the entire 10,787-instance `KGQAGen-10k` revealed 11 defective cases shown Table 6, a rate of error of 3.6%. Although this figure is relatively low, these instances expose recurring weaknesses in the generation and verification pipeline that warrant attention. The issues fall into three broad categories: self-answering prompts, hallucinated or incomplete relations, and errors inherited from the source knowledge graph.

The first issue, **self-answering questions**, was evident in items 4555 and 6931, where the question text directly includes the target answer. For example, asking about a subclass that 'has the same meaning as the English-language word 'city' leaves little ambiguity about the expected answer. Because our current verification process only checks that the SPARQL query returns at least one result overlapping the proposed answer set, it overlooks this form of lexical leakage and accepts the examples as valid.

The second and more prevalent category involves **hallucinated or incomplete knowledge**. In six cases (IDs 8318, 1105, 1164, 1529, 1825, and 10469), the model generated questions based on nonexistent or incoherent relationships, such as attributing architectural roles to historical political figures. In these situations, the LLM still produces syntactically valid queries, sometimes by leveraging loosely related property paths, allowing the verifier's overlap check to pass despite clear semantic errors. In a complementary failure mode, item 2297 exemplifies incomplete annotations: While multiple mathematicians satisfy the described criteria, only one is listed in the answer set. Since the verifier stops when it finds a match, it does not detect incompleteness.

A third source of error originates not from the generation process but from the **underlying knowledge graph itself in Wikidata [63]**. Items 9572 and 2046 illustrate this point: one references an entity mistyped as a product, the other includes an unlabeled identifier. Our pipeline implicitly treats Wikidata typing and labeling as authoritative, so these issues remain undetected unless caught during manual review.

The major cause of these verification failures lies in the limited scope of the current safeguard. The answer veridation and refinement (detailed in Section 4.3) of our `KGQAGen` checks are whether the SPARQL query compiles and whether its result set overlaps the proposed answer. Although effective in filtering out broken or irrelevant queries, this approach does not account for key semantic and structural issues. It does not detect leakage of lexical answers, does not require precise predicate alignment, does not check the joint coherence of query constraints, and does not validate type or label accuracy within the knowledge graph.

To address these gaps and further improve the quality of the data set beyond the current validation rate of 96.3%, we plan a series of targeted upgrades. First, we will implement a lexical filter to reject questions that contain their own answers. Second, we will enforce stricter predicate and type constraints within the generated queries to check against hallucinated relations. Third, we will introduce answer completeness audits through closure tests that ensure the full set of valid answers is captured. Finally, we will cross-check critical entity labels and types against alternative KG snapshots to catch inconsistencies and improve robustness across knowledge versions.

Table 6: `KGQAGen-10k` Error Analysis: Each case is displayed with concise issue diagnosis.

| Field | Content |
|---|---|
| **ID** | 4555 |
| **Question** | What subclass of 'city or town' has both a GND ID and is identified as the same concept as the English-language word 'city'? |
| **Answer** | city |
| **Issue** | **Self-answering:** The question is trivial; the answer is explicitly stated in the question itself. |
| **ID** | 8318 |
| **Question** | Who is the architect of Estadio Nacional de Costa Rica that was also a successful candidate in multiple Costa Rican general elections? |
| **Answer** | Ricardo Jiménez Oreamuno |
| **Issue** | **Hallucinated knowledge:** The question implies an architect relationship that does not exist; Ricardo Jiménez Oreamuno was a politician, not an architect. |
| **ID** | 1105 |
| **Question** | Which person who has been an owner of the Shroud of Turin is not a human individual, but rather a dynastic house? |
| **Answer** | Geoffroi de Charny, House of Savoy, Jeanne de Vergy, pope |
| **Issue** | **Hallucinated knowledge:** Geoffroi de Charny and House of Savoy are individuals, not dynastic houses. |
| **ID** | 1164 |
| **Question** | Which person, who held citizenship in the Ming, Qing, and short-lived Zhou dynasties, was both the father of Wu Yingxiong and the spouse of both Chen Yuanyuan and Empress Zhang, and led a revolt known as the Revolt of the Three Feudatories? |
| **Answer** | Kangxi Emperor, Kingdom of Tungning, Qing dynasty, Wu Sangui, Zheng Jing |
| **Issue** | **Hallucinated knowledge:** Zheng Jing is not the spouse of Chen Yuanyuan. |
| **ID** | 2297 |
| **Question** | Which mathematician both lent his name to the principle maintained by WikiProject Mathematics and is explicitly named as its discoverer or inventor? |
| **Answer** | Johann Peter Gustav Lejeune Dirichlet |
| **Issue** | **Hallucinated knowledge:** Many mathematicians have principles named after them; the annotation is incomplete and not unique. |
| **ID** | 1825 |
| **Question** | Which concept, characterized as 'unusual', is named after 'unusual', is the main subject of an entity named after 'unusual', and is also cited as a partially coincident concept by 'rarity'? |
| **Answer** | frequency, scarcity, unusual |
| **Issue** | **Hallucinated knowledge:** None of the ground truths are the subject of "World's Weirdest Animals". Its main subject is "creature". |
| **ID** | 10469 |
| **Question** | Which musical artist is associated with the genre that is said to be the same as "vintage" and has also performed in the genre represented by 'retro style'? |
| **Answer** | Adam Tsarouchis, Ahmad Bersaudara, Anna Jantar, Gloomwood, Nina Shatskaya, Type-B, VCTR-SCTR, Wieczór na dworcu w Kansas City |
| **Issue** | **Hallucinated knowledge:** Wieczór na dworcu w Kansas City is not a musical artist, but a retro style song. |
| **ID** | 1529 |
| **Question** | Which artist created a work that depicts the astronomical event discovered by Pierre Gassendi, and has as its main subject the transit of Mercury? |
| **Answer** | Mercury Passing Before the Sun |
| **Issue** | **Hallucinated knowledge:** Mercury Passing Before the Sun is the artwork, not the artist. The artist is Giacomo Balla. |
| **ID** | 6931 |
| **Question** | Which type of underwater vehicle is classified as both a subclass of submersible and shares this property with bathyscaphe, narco-submarine, and Osprey-class submersible? |
| **Answer** | Osprey-class submersible, bathyscaphe, bathysphere, narco-submarine, submersible drilling rig |
| **Issue** | **Self-answering:** The answer is trivially the same as the question's subject; provides no substantive challenge. |
| **ID** | 9572 |
| **Question** | Which company has produced a product used for flow measurement that includes a flow meter as a part? |
| **Answer** | Sage Metering |
| **Issue** | **Wikidata Mislabel:** The product of Sage Metering is labelled as "flow measurement" on Wikidata, but "flow measurement" is a task, not a product. |
| **ID** | 2046 |
| **Question** | Which tool that is a subclass of both 'physical tool' and is connected with 'level staff', is in turn a subclass of something that has the shape of a cylinder and is different from 'Rod'? |
| **Answer** | "Q9397141" |
| **Issue** | **Wikidata Mislabel:** The entity "Q9397141" doesn't contain the natural language label. |

# G  Experimental Details

This section provides comprehensive details about our experimental setup, including model configurations, training protocols, and evaluation procedures.

## G.1  Model Specifications

We evaluate three categories of models on `KGQAGen-10k`: (1) **Pure Language Models**, (2) **KG-RAG Systems**, and (3) **LLMs with Supporting Graphs**. These categories reflect increasing levels of external knowledge integration, ranging from purely parametric reasoning to symbolic augmentation and perfect-evidence setups.

**Pure Language Models** These models rely entirely on their internal parametric memory and are evaluated in a zero-shot setting, without any KG subgraph or retrieval. Their performance reflects inherent reasoning capability, factual coverage, and generalization.

- **LLaMA-3.1-8B-Instruct** [19]: An 8B instruction-tuned model from Meta's LLaMA 3.1 series. It is optimized for following task-specific instructions and shows improved reasoning performance compared to earlier LLaMA versions.

- **LLaMA2-7B** [61] : A general-purpose 7B model trained on publicly available data, serving as a foundational open-weight baseline for reasoning without instruction tuning.

- **Mistral-7B-Instruct-v0.2** [2]: An instruction-following model based on Mistral-7B with a 32k context window and standard attention. It is designed for accurate and efficient long-context reasoning.

- **GPT-4o-mini** [1]: A compact version of GPT-4o offering reduced latency and strong language understanding performance, suitable for real-time applications.

- **GPT-4** [1]: OpenAI's flagship model known for robust multi-step reasoning, long-context understanding, and generalization across a wide array of tasks.

- **DeepSeek-Chat** [11]: A dialogue-oriented LLM developed by DeepSeek and fine-tuned for task completion and conversational fluency aligned with human feedback.

- **GPT-4o** [1]: A unified multimodal model capable of handling text, image, and audio inputs. We use it in a text-only setup to assess its advanced reasoning capabilities.

- **GPT-4.1** [41]: An updated variant of GPT-4 that improves long-context performance, factual grounding, and consistency in complex prompt execution.

**KG-RAG Systems** Knowledge Graph Retrieval-Augmented Generation (KG-RAG) systems incorporate structured symbolic evidence from a KG to assist reasoning and answer generation. These models access retrieved subgraphs at runtime and vary in how they integrate retrieved content—either as conditioning input or through decoding constraints.

- **RoG (LLaMA2-7B)** [33]: Fine-tuned on `KGQAGen-10k`'s training split, using annotated supporting subgraphs to supervise faithful reasoning path generation for answer and explanation prediction.

- **GCR (LLaMA-3.1 + GPT-4o)** [34]: Fine-tuned its path generation module with supporting subgraphs, leveraging a KG-Trie to constrain decoding and using GPT-4o for final answer synthesis.

- **ToG (GPT-4o)** [57]: Adapted to Wikidata by replacing Freebase API calls with SPARQL queries and evaluated zero-shot, interactively exploring the KG and generating answers from the retrieved subgraph without fine-tuning or parameter adjustment.

- **PoG (GPT-4o)** [9]: Applied as a zero-shot prompting-based agent that dynamically decomposes, explores, and self-corrects over Wikidata for each test question, without any dataset-specific fine-tuning.

**LLM with Supporting Subgraph**    To estimate the upper bound of KG-augmented QA performance, we provide models with the gold supporting subgraph used during data generation. This simulates a perfect-retrieval setting, where the model receives all and only the minimal evidence needed to answer correctly. These experiments assess whether models can effectively reason over structured KG input when retrieval is assumed to be ideal.

- **LLaMA2-7B (w/ SP)** [61]: The model is provided with the gold subgraph and asked to generate the answer. This tests the reasoning capacity of a smaller open-weight model under ideal symbolic input.
- **GPT-4o (w/ SP)** [1]: The same setup as above, but with GPT-4o as the base model. This configuration reflects an upper-bound for KG-RAG systems when both retrieval and reasoning are ideal.

### G.2    Beyond Exact Match: Introducing LASM

While the limitations of exact match evaluation in KGQA are well-recognized [50, 64], few works have proposed principled solutions. To address this gap, we introduce LLM-Assisted Semantic Match (LASM), a novel evaluation scheme that goes beyond surface-level equivalence by leveraging the semantic understanding capabilities of large language models.

The core idea of LASM is to use an LLM verifier to assess semantic similarity between predicted and ground truth answers. When a model's prediction fails the exact string match, LASM invokes a GPT-4o-mini judge to determine whether the prediction is semantically equivalent to the gold answer. This approach enables LASM to properly credit models for generating meaningfully correct responses that traditional metrics would overlook due to syntactic or lexical variation. To quantify the impact of LASM, we compare model performance on the FreebaseQA dataset [23] under both exact match and LASM evaluation. As shown in Table 7, LASM yields substantial improvements across all key metrics, including accuracy (+5.3%), Hit@1 (+5.3%), and F1 (+5.0%). These gains demonstrate the effectiveness of semantic matching in capturing valid model predictions that exact match misses.

| Scoring | Accuracy | Hit@1 | F1 | Precision | Recall |
|---|---|---|---|---|---|
| Exact Match | 90.39 | 90.39 | 88.08 | 87.08 | 90.39 |
| LASM | 95.72 | 95.67 | 93.12 | 92.04 | 95.65 |

Table 7: FreeBaseQA results with GPT-4o. LASM consistently recovers semantically correct predictions missed by exact match, leading to substantial metric improvements.

Beyond offering a more robust and nuanced assessment of model outputs, LASM has important implications for the development and evaluation of KGQA systems. By rewarding models for semantic correctness rather than rigid string matching, LASM promotes the development of systems that prioritize meaning over surface form. Moreover, as a fully automated method that does not rely on dataset-specific rules or annotations, LASM is readily applicable to any KGQA benchmark, enabling more meaningful cross-dataset comparisons.

To ensure evaluation reliability, LASM is applied only when the prediction fails the exact match, serving as a selective fallback rather than a replacement for EM. This design mitigates potential risks of LLM-based scoring such as hallucinations or factual drift, as LASM is invoked for only a small subset of cases. To assess its accuracy, we manually inspected 100 randomly sampled instances where LASM was triggered. Two human reviewers independently verified semantic correctness using Wikipedia, identifying only one false positive (1% error rate)—a case where the prediction *"Austronesian languages"* was judged equivalent to the ground truth *"Oceanic"*. This low error rate demonstrates that LASM provides a reliable supplement to exact match evaluation. Furthermore, similar approaches in the NLG domain support the robustness of LLM-based evaluation, as shown by recent studies such as G-Eval [31], FActScore [37], MT-Bench [80], and QAFactEval [68], which report strong alignment between GPT-4-based judgments and human evaluation.

In summary, LASM represents a principled and generalizable approach to overcoming the limitations of traditional exact match evaluation in KGQA. By incorporating semantic awareness through LLM-based similarity judgments, LASM provides a more reliable and nuanced assessment of model performance, paving the way for the development of more robust question answering systems. As

we will demonstrate through extensive experiments in Section 5, LASM offers a valuable tool for evaluating and advancing the state of the art in KGQA.

### G.3 Experimental Setup

We evaluate all models on `KGQAGen-10k` using a standardized split of 8,629/ 1,079/1,079 train/e-val/test. For KG-RAG systems, we adapt each model to work with Wikidata by replacing their original knowledge base interfaces with SPARQL queries to the Wikidata endpoint.

**Training and Inference Protocols**  RoG [33] employs a planning-retrieval-reasoning pipeline where LLaMA2-7B first generates candidate relation paths, which are then matched against the knowledge graph using constrained breadth-first search. The retrieved reasoning paths, combined with the original question, guide the model to generate both answers and explanations. We fine-tune the entire pipeline on our training split, using the supporting subgraphs from dataset construction as supervision for faithful path generation.

GCR [34] enforces graph faithfulness through constrained decoding. Prior to inference, we construct a KG-Trie index that efficiently captures all valid reasoning paths within a fixed hop limit. During generation, a fine-tuned LLaMA-3.1 model produces candidate paths under strict KG-Trie constraints, ensuring only valid graph traversals. These candidates are then passed to GPT-4o for inductive reasoning and answer synthesis. Similar to RoG, we leverage our supporting subgraphs for training the path generation component.

In contrast, ToG [57] and PoG [9] operate without fine-tuning, treating the LLM as an agent that interactively explores the knowledge graph. ToG constructs reasoning trees by iteratively selecting relations and entities based on question semantics, while PoG enhances this with adaptive planning and self-correction mechanisms. For both models, we implement direct Wikidata integration, allowing them to dynamically query the knowledge base during inference without dataset-specific training.

**Evaluation Metrics**  We evaluate each KGQA system using 4 complementary Hit@1, Precision, Recall, and F1—under two answer-matching schemes: Exact Match (EM) and LASM. **EM** considers a prediction correct only if the model's answer set exactly matches the ground-truth set after basic normalization, which includes lowercasing and alphabetically sorting answers. This is a strict string-level comparison that does not account for synonyms, paraphrases, or other forms of semantic equivalence. Hit@1 measures whether the model's top-ranked answer appears in the ground-truth set. Precision, Recall, and F1 capture the degree of set overlap: Precision reflects the proportion of predicted answers that are correct, Recall captures the proportion of ground-truth answers that are retrieved, and F1 is their harmonic mean—together highlighting whether a model tends to over- or under-generate. **LASM** extends this evaluation by replacing literal comparison with a GPT-4o-mini verifier that determines whether the predicted and ground-truth answers are semantically equivalent. We then recompute all five metrics based on this semantic agreement. This two-tiered protocol offers a comprehensive view of model performance, balancing surface-level exactness with meaning-level correctness.

**Cross-Dataset Evaluation.**  To further contextualize model performance, we perform a cross-dataset comparison between our proposed `KGQAGen-10k` and two widely used KGQA benchmarks, WebQSP [73] and CWQ [58]. We evaluate four representative frameworks—RoG [33], GCR [34], ToG [57], and PoG [9]—under consistent inference settings. For WebQSP and CWQ, we report results from the respective original papers, while for `KGQAGen-10k`, we re-run each model using identical retrieval and reasoning configurations.

As shown in Table 8, all models achieve high accuracy on traditional benchmarks (e.g., RoG reaches 85.7 on WebQSP) but experience a substantial drop on `KGQAGen-10k`, where the same model attains only 21.3. This contrast highlights the increased reasoning complexity and stricter grounding requirements in our dataset. Moreover, the relative ranking of models remains consistent, but the gaps between them widen on `KGQAGen-10k`, indicating that it better differentiates model capabilities under more realistic and verifiable evaluation conditions.

| Model | WebQSP (Hit@1) | CWQ (Hit@1) | KGQAGen-10k (Hit@1) |
|---|---|---|---|
| RoG | 85.7 | 62.6 | 21.3 |
| GCR | 92.2 | 75.8 | 52.5 |
| ToG | 82.6 | 67.6 | 52.6 |
| PoG | 87.3 | 75.0 | 54.0 |

Table 8: Cross-dataset comparison of KGQA performance. Results on WebQSP and CWQ are taken from the respective original papers; `KGQAGen-10k` results are obtained under identical evaluation protocols.

