# OpenReview forum: "Diagnosing and Addressing Pitfalls in KG-RAG Datasets: Toward More Reliable Benchmarking"
_NeurIPS.cc/2025/Datasets_and_Benchmarks_Track — NeurIPS 2025 Datasets and Benchmarks Track poster_

### Official Review · Reviewer_3ELu · 2025-06-17

**Rating:** 5
**Confidence:** 4

**Summary:**

This paper identifies significant quality issues in existing Knowledge Graph Question Answering (KGQA) benchmarks, such as inaccurate annotations and ambiguous questions. To address these problems, the authors propose KGQAGen, an LLM-in-the-loop framework that generates high-quality, challenging, and verifiable QA instances by combining structured knowledge grounding, LLM-guided generation, and symbolic verification. Using this framework, they create KGQAGen-10k, a large-scale benchmark based on Wikidata. Experiments show that current state-of-the-art KGQA models perform poorly on this new benchmark, underscoring the need for more rigorous evaluation datasets and demonstrating the effectiveness of KGQAGen for advancing KGQA research.

**Additional Feedback:**

Please check the feedback in the weakness section.

**Dataset Code Accessibility:**

Yes

**Dataset Code Comments:**

I have checked the dataset code, and they are accessible.

**Ethical Comments:**

I don't fine significant ethical concerns in the paper.

**Ethical Considerations:**

No, there are no or only very minor ethics concerns

**Final Justification:**

Since I have already given a positive review, good luck for this paper!

**Limitations Weaknesses:**

I strongly suggest that the authors combine the main body and the appendix into a single file, as I almost missed the appendix otherwise.

I suggest the authors to check another paper which seems to focus on the similar topic: "Can Knowledge Graphs Make Large Language Models More Trustworthy? An Empirical Study over Open-ended Question Answering". Even the benchmarks size is smaller than this work and mainly focus on open-ended QA, the idea is still similar (like using LLM-guided iterative generation, seed subgraph, etc.) and the authors may consider to mention it in the related work section.

**Strengths Contributions:**

The motivation of the paper is quite clear: as the previous benchmarks like webqsp and cwq have the quality issues and most of the samples are out-dated.

The proposed benchmark seems to be much harder compared to the existing ones and help to expose the limitations of the current models.

I think the manual auditing for the existing KGQA benchmarks in Appendix B is valuable and useful.

Overall, I think this is a good paper and seems natural to me. The writing and experiments are all good.

---

> ### Author Rebuttal · Authors · 2025-07-31
>
> 1. Thank you for the suggestion. If our paper is accepted, we will combine the main paper and appendix into a single file for the camera-ready version to improve accessibility and ensure that no important content is overlooked.
> 2. Thank you for bringing this paper to our attention. We have carefully reviewed “Can Knowledge Graphs Make Large Language Models More Trustworthy? An Empirical Study over Open-ended Question Answering” (Sui et al., ACL 2025) and agree that it is highly relevant to our work. OKGQA [1] is a valuable effort to assess whether knowledge graphs reduce hallucinations in open-ended, non-factoid QA, including a perturbed variant (OKGQA-P) to test robustness under KG noise. While both OKGQA and our work use template-based question generation and subgraph extraction, their focus is on paragraph-style responses evaluated with hallucination metrics, whereas we generate SPARQL-validated QA pairs for structured, multi-hop KGQA. Their use of static PPR-based sampling contrasts with our iterative LLM-guided expansion, which better controls complexity and ensures answer correctness. We will cite OKGQA as a complementary approach addressing trustworthiness, while our framework emphasizes symbolic precision and scalability.
>
> [1] Sui, Yuan, et al. "Can knowledge graphs make large language models more trustworthy? an empirical study over open-ended question answering." arXiv preprint arXiv:2410.08085 (2024).

---

> > ### Author Response · Authors · 2025-08-04
> >
> > We appreciate your prompt acknowledgment. Thank you once again for your valuable feedback and guidance throughout the review process.

---

> > > ### Comment · Reviewer_3ELu · 2025-08-05
> > >
> > > Since I have already given a positive review, good luck for this paper!

---

> > > > ### Author Response · Authors · 2025-08-05
> > > >
> > > > Appreciate your support and encouragement! Thank you again.

---

### Official Review · Reviewer_Hp4s · 2025-06-20

**Rating:** 4
**Confidence:** 4

**Summary:**

Considering that the current widely used Knowledge Graph Question Answering (KGQA) benchmarks have low quality issues, such as inaccurate ground-truth labels, outdated or incomplete answers, and poorly constructed or ambiguous questions, this work proposes KGQAGen, a modular, scalable framework that uses an LLM-in-the-loop approach for generating verifiable and challenging QA instances grounded in Wikidata. They also release KGQAGen-10k, a high-quality benchmark constructed with their framework, and show that even state-of-the-art LLMs and KG-RAG models struggle with it.

**Additional Feedback:**

Could the authors provide more insights into failure cases where SPARQL validation fails or revisions do not converge?

**Dataset Code Accessibility:**

Yes

**Ethical Considerations:**

No, there are no or only very minor ethics concerns

**Final Justification:**

The authors have addressed some of my concerns in the rebuttal. I think my original decision is fair. Therefore, I maintain my score.

**Limitations Weaknesses:**

1. The results seem to be obtained by single-run evaluations, without error bars or statistical tests.
2. The quality of the generated dataset partially depends on LLM outputs. This may introduce bias to the dataset. Although symbolic verification may mitigate hallucination, the paper could elaborate more on how different LLMs impact the dataset.
3. Only wikidata is used for symbolic grounding. It would be better to verify the data construction approach on more real-world KGs.

**Strengths Contributions:**

1. This paper is well-written and easy to follow.
2. The authors highlight a pressing and under-addressed issue in KGQA evaluation: the low quality of many standard benchmarks. Their systematic audit brings needed attention to this problem.
3. KGQAGen is a well-designed framework that ensures factual correctness and supports multi-hop reasoning.
4. The released dataset is diverse, challenging, and well-validated (e.g., manual evaluation reports a 96% correctness rate)
5. A detailed evaluation across multiple model categories (pure LLMs, KG-RAG models, and LLMs with perfect retrieval) has been conducted. Furthermore, the authors also introduce the LLM-Assisted Semantic Match (LASM) mechanism to go beyond exact-match limitations.

---

> ### Author Rebuttal · Authors · 2025-07-31
>
> 1.Thanks for the advice.  In the KG-RAG literature [1–7], most evaluations are conducted using single-run results, and we follow this convention for our baseline models. To verify the robustness of such evaluation setting, we further conduct a multiple run experiments. We reran the two baseline models (RoG and GCR) three times with different random seeds. The resulting standard deviations are ≤ 0.45 pp for RoG and ≤ 0.14 pp for GCR across all metrics, demonstrating near-deterministic behavior. This explains why prior work (and our original submission) reported single runs without error bars.
>
> | Metric    | RoG (mean ± σ)   | GCR (mean ± σ)   |
> |-----------|------------------|------------------|
> | Accuracy  | 20.58 ± 0.37     | 49.54 ± 0.12     |
> | Hit@1     | 21.72 ± 0.31     | 52.61 ± 0.12     |
> | F1        | 18.01 ± 0.34     | 49.30 ± 0.06     |
> | Precision | 17.86 ± 0.30     | 50.64 ± 0.06     |
> | Recall    | 20.58 ± 0.37     | 49.54 ± 0.12     |
>
>
> 2.Thank you for the comment. We agree that the choice of LLM impacts the generated dataset and may introduce bias. We use GPT-4.1 as our primary model because it is currently among the most capable publicly available LLMs for factuality, reasoning, and semantic understanding, which are crucial for generating high-quality and diverse QA pairs. GPT-4.1 has been shown to outperform smaller or earlier models in both accuracy and alignment with human preferences in various evaluation settings.
>
> To assess the impact of different LLMs on generation quality, we conduct an ablation study by replacing the default generation model with alternative LLMs. Specifically, we compare LLaMA-3.1-8B-Instruct and GPT-4o-mini using slice of 100 seed entities from Wikidata. Each variant is evaluated based on (1) generation success rate, the proportion of seeds that yield valid questions; (2) end-to-end success rate, the proportion that pass SPARQL validation.
>
> - LLaMA-3-70B generates 57 questions (57% generation rate), of which 35 pass validation (35% end-to-end success, 61% validation rate on generated questions). The model often fails to preserve multi-hop structure, frequently simplifying prompts into 1-hop facts or hallucinating relations.
>
> - GPT-4o-mini performs more reliably, producing 65 questions (65% generation rate) and validating 41 (41% end-to-end success, 63% validation rate). GPT-4o-mini better retains triple structure and handles simple multi-hop chains, but still underperforms on more complex reasoning. These results highlight the importance of LLM capacity and structure-aware generation, as the full KGQAGen pipeline with GPT-4.1 achieves the highest performance across all metrics.
>
> In contrast, our full pipeline with GPT-4.1 generates 82 questions and validates 77 of them, achieving a 94% success rate on generated questions and a 77% end-to-end success rate. These results show that both model capacity and our structure-aware guidance materially affect outcomes, while symbolic SPARQL verification mitigates hallucinations and constrains bias.
>
> 3.Thank you for the helpful suggestion. We chose Wikidata [9] as the starting point because it provides a practical and well-supported environment for developing and evaluating our KGQAGen framework. Its large scale (over 100 million entities and 1.7 billion triples) allows for diverse relation paths and reasoning patterns. The rich schema—with qualifiers, temporal information, and statement ranks, supports more complex and realistic question types. It also offers a stable public SPARQL endpoint, which makes all queries easily reproducible, and its CC0 license allows for open sharing of the generated datasets.
>
> However, our approach is not limited to Wikidata KG. To demonstrate generality, we applied KGQAGen to DBpedia [8] using the same models and prompts, requiring only minor modifications to the SPARQL interface and a lightweight schema adapter. On a randomly selected 100-seed sample, KGQAGen generated 74 candidate questions and validated 61 end-to-end with SPARQL. The lower validation rate compared to Wikidata is primarily due to DBpedia’s predicate heterogeneity and the prevalence of text-valued infobox fields, which introduce ambiguity and larger answer sets. Nevertheless, our symbolic verification module remained effective in filtering noisy generations, and the modest adapter changes were sufficient for high-quality output.
>
> In sum, the approach is not tied to Wikidata: given a SPARQL endpoint (online or local deployed) and a short schema/label adapter, KGQAGen transfers with minimal effort. We will include our DBpedia adapter and the 100-seed results in the revision, and plan to release adapter templates to further substantiate cross-KG generality.
>
> 4.Thanks for the additional feedback. We analyzed 420 instances filtered out by our SPARQL-based answer validation. The main dumped types include: SPARQL queries returning empty results (166 cases, 39.5%), queries yielding excessively large answer sets (184 cases, 43.8%), and answer mismatches where the annotated answer was not present in the SPARQL results or did not match (70 cases, 16.7%). These issues are primarily due to incomplete or ambiguous generated Q-A, limitations in KG coverage, or challenges in query formulation. In future work, we will enhance our validation module by adding tighter constraints, clarification prompts, and detailed error classification. This will improve both explainability and our ability to systematically address specific types of validation failures.
>
> [1] Luo, Linhao, et al. "Reasoning on graphs: Faithful and interpretable large language model reasoning." arXiv preprint arXiv:2310.01061 (2023).
>
> [2] Sun, Jiashuo, et al. "Think-on-graph: Deep and responsible reasoning of large language model on knowledge graph." arXiv preprint arXiv:2307.07697 (2023).
>
> [3] Chen, Liyi, et al. "Plan-on-graph: Self-correcting adaptive planning of large language model on knowledge graphs." Advances in Neural Information Processing Systems 37 (2024): 37665-37691.
>
> [4] Luo, Linhao, et al. "Graph-constrained reasoning: Faithful reasoning on knowledge graphs with large language models." arXiv preprint arXiv:2410.13080 (2024).
>
> [5] Mavromatis, Costas, and George Karypis. "Gnn-rag: Graph neural retrieval for large language model reasoning." arXiv preprint arXiv:2405.20139 (2024).
>
> [6] Jiang, Jinhao, et al. "Structgpt: A general framework for large language model to reason over structured data." arXiv preprint arXiv:2305.09645 (2023).
>
> [7] Xiong, Guanming, Junwei Bao, and Wen Zhao. "Interactive-kbqa: Multi-turn interactions for knowledge base question answering with large language models." arXiv preprint arXiv:2402.15131 (2024).
>
> [8] Lehmann, Jens, et al. "Dbpedia–a large-scale, multilingual knowledge base extracted from wikipedia." Semantic web 6.2 (2015): 167-195.
>
> [9] Vrandečić, Denny, and Markus Krötzsch. "Wikidata: a free collaborative knowledgebase." Communications of the ACM 57.10 (2014): 78-85.

---

> > ### Comment · Reviewer_Hp4s · 2025-08-05
> >
> > Thanks for the detailed response. After reading the rebuttal context, I think my original decision is fair. Therefore, I plan to maintain my original scores. BTW, something could be better: most citations you used in the rebuttal context are from Arxiv, but I know that some of them have been published by venues. I suggest the author cite their officially published version next time.

---

> > > ### Author Response · Authors · 2025-08-05
> > >
> > > Thank you for taking the time to review our rebuttal and share your thoughts. We appreciate your suggestion regarding the citations and will make sure to update them with the officially published versions in the final submission.

---

> ### Author Response · Authors · 2025-08-04
>
> Thanks again for your thoughtful feedback and guidance throughout the review process.
> We just wanted to follow up regarding our submitted rebuttal. As the discussion period closes soon (by August 6th, AoE), we would greatly appreciate the opportunity to clarify or address any remaining questions or concerns you might have. If there is anything further we can provide, we are more than happy to do so.
> If everything looks good on your end, we would be truly grateful if you would consider raising the score. We sincerely appreciate your support.

---

### Official Review · Reviewer_UW2q · 2025-07-03

**Rating:** 4
**Confidence:** 3

**Summary:**

This paper conducts a comprehensive study on Knowledge Graph-based Question Answering (KGQA). It systematically audits 16 prevalent KGQA datasets, identifying crucial quality and evaluation flaws. To address these issues, the authors introduce KGQAGen, a scalable framework guided by large language models (LLMs) to generate challenging, evidence-based, and verifiable KGQA benchmarks. Leveraging this framework, they construct the KGQAGen-10k dataset, a 10,000-example benchmark that analyzes question characteristics and evaluates various KG-RAG models, highlighting significant performance disparities and areas for future enhancement.

**Additional Feedback:**

Could you provide experimental evidence showing that the LASM metric offers a reliable assessment of model performance (e.g., its consistency with human evaluations), or cite relevant references (such as other evaluation methods based on LLMs) that support the validity of this metric?

**Dataset Code Accessibility:**

Yes

**Dataset Code Comments:**

The authors have released the code of KGQAGen at https://github.com/liangliang6v6/KGQAGen and made the KGQAGen-10k dataset publicly available at https://huggingface.co/datasets/lianglz/KGQAGen-10k.

**Ethical Considerations:**

No, there are no or only very minor ethics concerns

**Final Justification:**

Authors' response has solved my concerns, so I decide to keep my score unchanged.

**Limitations Weaknesses:**

L1: As noted by the authors in the Limitations section, KGQAGen is still constrained by the underlying knowledge graph. Although the authors mention that existing KGQA datasets may suffer from the issue of "outdated answers", KGQAGen does not appear to incorporate specific mechanisms to address this problem in its dataset generation process.

L2: The paper introduces the LLM-Assisted Semantic Match (LASM) metric, but it does not seem to explain why this LLM-based evaluation metric is reasonable or how it can accurately reflect model performance.

**Strengths Contributions:**

S1: This paper has paid attention to the potential errors or ambiguities in existing KGQA datasets and conducted large-scale manual verification, which has important potential implications for research in this field.

S2: The proposed KGQAGen framework introduces a novel and effective methodology for generating high-quality KGQA datasets from existing knowledge graphs with limited human effort. The framework provides a feasible solution for dataset construction in the KGQA domain.

S3: The paper is well-organized and clearly written, with illustrative examples provided for key technical components, facilitating a better understanding of the proposed methodology and its contributions.

---

> ### Author Rebuttal · Authors · 2025-07-31
>
> L1 Response:
>
> Thank you for this feedback. While rely on the underlying knowledge graphs, KGQAGen indeed provides a way to keep the questions up-to-date. Modern KGs like Wikidata are continuously evolving, and our dataset includes the associated SPARQL queries for each question. These queries can be re-executed on the latest version of the KG to update the answer set accordingly, ensuring the answer up-to-date. As noted in Line 302 (Page 7), our framework uses SPARQL queries that let datasets be "periodically revalidated as the KG evolves—making KGQAGen-generated benchmarks both accurate at creation and maintainable over time."
>
>
> L2 Response with feedback:
>
> Thank you for this important question about LASM's reliability. We introduced LASM to address a key problem in KGQA evaluation: exact match often misses semantically correct answers due to different word forms, abbreviations, or entity names (e.g., "AUD" vs. "Australian dollar"). We fully understand the concerns regarding the reliability of LLM-based evaluation, including potential hallucinations or knowledge gaps. To mitigate these risks, LASM is applied only when exact match fails. In other words, it is used for a limited portion of the dataset—specifically, when the predicted answer does not exactly match the ground truth. This selective application ensures that LASM functions as a fallback mechanism rather than replacing exact match as the primary evaluation metric (as noted in Line 344, Page 8).
>
> To test LASM's accuracy, we manually checked 100 random cases where exact match failed and LASM was used. Two human reviewers independently checked each case using Wikipedia to verify if the predicted answer was semantically correct. LASM had only one false positive out of 100 cases (1% error rate). In this instance (ID 4566), the prediction was ['Austronesian languages'], while the ground truth was ['Oceanic']. Although the two terms are related, they are not semantically equivalent, and LASM incorrectly judged them as matching. Since LASM only applies to a small subset of predictions where exact match fails, this low error rate makes it a reliable supplement to exact match evaluation.
>
> We will include and discuss more LLM-based evaluation methods in the related work section. For example, in the natural language generation (NLG) domain, it is challenging to automatically evaluate the quality of generated text, and human evaluation is often costly and time-consuming. As a result, using LLMs as evaluators for long-form responses has become the de facto standard [1]. Studies show GPT-4 evaluation matches human judgment well, with G-Eval [2] achieving 0.51 Spearman correlation on summarization tasks and FActScore [3] matching human scores within 2% error. And more large-scale studies like MT-Bench [4] and QAFactEval [5] also prove the reliability of factual consistency evluation with LLM-assistant.
>
> [1] Gu, Jiawei, et al. "A survey on llm-as-a-judge." arXiv preprint arXiv:2411.15594 (2024).
>
> [2] Liu, Yang, et al. "G-eval: NLG evaluation using gpt-4 with better human alignment." arXiv preprint arXiv:2303.16634 (2023).
>
> [3] Min, Sewon, et al. "Factscore: Fine-grained atomic evaluation of factual precision in long form text generation." arXiv preprint arXiv:2305.14251 (2023).
>
> [4] Zheng, Lianmin, et al. "Judging llm-as-a-judge with mt-bench and chatbot arena." Advances in neural information processing systems 36 (2023): 46595-46623.
>
> [5] Wu, Alexander R. Fabbri Chien-Sheng, and Wenhao Liu Caiming Xiong. "QAFactEval: Improved QA-Based Factual Consistency Evaluation for Summarization." (2023).

---

> ### Author Response · Authors · 2025-08-04
>
> Thanks again for your thoughtful feedback and guidance throughout the review process.
> We just wanted to follow up regarding our submitted rebuttal. As the discussion period closes soon (by August 6th, AoE), we would greatly appreciate the opportunity to clarify or address any remaining questions or concerns you might have. If there is anything further we can provide, we are more than happy to do so.
> If everything looks good on your end, we would be truly grateful if you would consider raising the score. We sincerely appreciate your support.

---

### Official Review · Reviewer_e6N6 · 2025-07-04

**Rating:** 5
**Confidence:** 4

**Summary:**

The paper addresses critical quality issues in existing KGQA datasets, such as inaccurate answers and ambiguous questions, which reveal the reliable evaluation of KG-RAG systems. To overcome these challenges, the authors introduce KGQAGen, a scalable, LLM-in-the-loop framework for generating high-quality, verifiable KGQA datasets grounded in structured knowledge from Wikidata. Using this framework, they create KGQAGen-10k, a challenging benchmark with well-grounded question-answer pairs.

**Dataset Code Accessibility:**

Yes

**Dataset Code Comments:**

The datasets and associated usage code are available and easy to use with Huggingface. Nice README document is provided on GitHub with clear instructions and environment requirements.

**Ethical Considerations:**

No, there are no or only very minor ethics concerns

**Final Justification:**

The paper offers substantial contributions to KG-RAG benchmarking and could benefit future studies in this area.

**Limitations Weaknesses:**

1. The authors could consider showing the performance difference between traditional graphQA datasets and the proposed ones, which would be interesting to see some contradictory and surprising observations.

2. Lack of ablation studies to systematically explain the framework design choices.

3. Limited scope of question types and reasoning complexity. Could consider introducing more quantitative measures of task/reason complexity with graph complexity, such as the method mentioned in [1]. [1] Zhu, Kaijie, et al. "Dyval: Dynamic evaluation of large language models for reasoning tasks." arXiv preprint arXiv:2309.17167 (2023).

**Strengths Contributions:**

1. A systematic manual audit revealing widespread quality issues in popular KGQA benchmarks, a lot of effort.

2. The authors develop KGQAGen, a modular framework leveraging LLMs and symbolic verification for scalable, high-quality dataset construction.

3. The authors further generate KGQAGen-10k, which exposes limitations in current KG-RAG models and emphasizes the importance of effective retrieval and reasoning mechanisms for future progress.

---

> ### Author Rebuttal · Authors · 2025-07-31
>
> 1.Thank you for the suggestion. To provide a more complete picture, we have included a performance comparison across WebQSP, CWQ, and our proposed KGQAGen-10k dataset, using results reported in the original works of RoG [1], ToG [2], PoG [3], and GCR [4].
>
> As shown in table below, while Hit@1 scores on WebQSP and CWQ appear relatively high across all models (e.g., RoG reaches 85.7 on WebQSP), performance on KGQAGen-10k is substantially lower for the same models, especially for RoG (21.3), revealing the increased reasoning complexity and stricter answer grounding in our benchmark. The performance gaps between models are also more pronounced on KGQAGen-10k, showing its value in differentiating KGQA capabilities under rigorous evaluation. Meanwhile, as our analysis in Section 3 shows, WebQSP and CWQ suffer from issues such as outdated facts, annotation noise, and ambiguous question formulations, which can lead to inflated or unreliable performance estimates. These results confirm that KGQAGen-10k provides a more challenging and diagnostic evaluation setting for modern KGQA models.
>
> | Model | WebQSP Hit@1 | CWQ Hit@1 | KGQAGen-10k Hit@1 |
> |-------|--------------|-----------|-------------------|
> | RoG   | 85.7         | 62.6      | 21.3              |
> | GCR   | 92.2         | 75.8      | 52.5              |
> | ToG   | 82.6         | 67.6      | 52.6              |
> | PoG   | 87.3         | 75.0      | 54.0              |
>
>
> 2.We conduct an ablation study to examine our design choices. In our pipeline, questions are generated through iterative LLM-guided subgraph expansion. To further evaluate the impact of different components, we test both random subgraph selection and alternative LLMs. Additionally, we compare results with and without the SPARQL-based answer validation to assess its effect on overall quality.
>
> Specifically, we use 100 randomly selected seed entities to generate and validate results under three settings:
> - A1: Random subgraph selection (replaces LLM-guided selection with uniform random sampling from the local neighborhood).
> - A2: LLM model variation, generation model replaced with Llama-3.1-8B-Instruct or GPT-4o-mini.
> - A3: With or without answer validation (SPARQL-based symbolic verification).
>
> Notably, A3 is not an independent setting, but rather a cross-setting evaluation axis: for each configuration (A1, A2, and the full pipeline), we present results both before and after applying SPARQL-based answer validation. This allows us to directly quantify the contribution of symbolic verification to overall data quality across all ablation variants. We report two metrics: the generation success rate (percentage of seeds that produce questions) and the end-to-end success rate (percentage resulting in fully validated QA pairs).
>
> - The full KGQAGen generates 82 questions and validates 77, achieving a 94% generation success rate and 77% end-to-end success rate.
>
> - A1 removes LLM guidance and selects a 2-hop subgraph by randomly choosing 2 neighbors. All 100 seeds produce questions, but only 62 pass SPARQL validation (62% end-to-end rate). Random subgraphs often include extra entities as possible answers, causing ambiguity. KGQAGen’s guided exploration and constraints narrow answer scope, showing the importance of subgraph selection and sufficiency modules.
>
> - A2 examines the impact of LLM choice. LLaMA-3-70B generates 57 questions (57% generation rate), with 35 passing SPARQL validation (35% end-to-end, 61% on generated questions). The model often loses triple-level details or reduces multi-hop to one-hop questions. In contrast, GPT-4o-mini performs better, generating 65 questions and validating 41 (65% and 41%, respectively, with 63% on generated questions). GPT-4o-mini better preserves structure and can generate multi-hop questions, but still struggles with complex subgraphs or ambiguous queries. These results indicate that while smaller models like GPT-4o-mini outperform LLaMA-3-70B in retaining structure, both are less effective than the full KGQAGen pipeline with GPT-4.1, which benefits from higher model capacity and structured guidance.
>
> 3.Thank you for the helpful suggestion. We have added a graph-based reasoning complexity analysis following the DyVal framework [5]. Specifically, we compute six metrics for each supporting subgraph—number of nodes, edges, average degree, depth, width, and extra links—to capture both the scale and structural complexity of reasoning. These metrics reflect aspects of reasoning difficulty such as multi-hop depth, breadth of intermediate steps, and presence of cycles or shortcuts.
>
> The table below summarizes statistics across all 10,787 examples. 98% of questions require 2–5 hops; 84% of graphs contain 5–30 entities; 83% have 4–28 relations; 79% have width between 4–20; and 92% are fully connected. This confirms our dataset includes diverse and non-trivial reasoning graphs matching the structured complexity targeted by DyVal.
>
> | Measure      | Distribution      | Percentage |
> |--------------|------------------|------------|
> | Depth        | 2–5 hops         | 98%        |
> | Nodes        | 5–30             | 84%        |
> | Edges        | 4–28             | 83%        |
> | Width        | 4–20             | 79%        |
> | Reachability | Fully connected  | 92%      |
>
>
> [1] Luo, Linhao, et al. "Reasoning on graphs: Faithful and interpretable large language model reasoning." arXiv:2310.01061 (2023).
> [2] Sun, Jiashuo, et al. "Think-on-graph: Deep and responsible reasoning of large language model on knowledge graph." arXiv:2307.07697 (2023).
> [3] Chen, Liyi, et al. "Plan-on-graph: Self-correcting adaptive planning of large language model on knowledge graphs." NeurIPS 37 (2024): 37665-37691.
> [4] Luo, Linhao, et al. "Graph-constrained reasoning: Faithful reasoning on knowledge graphs with large language models." arXiv:2410.13080 (2024).
> [5] Zhu, Kaijie, et al. "Dyval: Dynamic evaluation of large language models for reasoning tasks." arXiv:2309.17167 (2023).

---

> > ### Comment · Reviewer_e6N6 · 2025-08-04
> >
> > I thank the authors for incorporating the ablation studies, comparisons with traditional graph datasets, detailed explanations of model design choices, and quantitative evaluations on reasoning graphs. I have updated my score to 5. I hope the authors will include these results, along with further discussion/insights/explanations/conclusions (not currently in the rebuttal), in the final version.

---

> > > ### Author Response · Authors · 2025-08-04
> > >
> > > Thank you very much for taking the time to review our updates and for your feedback. We sincerely appreciate the score revision and your encouraging remarks. We will make sure to include the new results, along with expanded discussion and insights.

---

> ### Author Response · Authors · 2025-08-04
>
> Thanks again for your thoughtful feedback and guidance throughout the review process.
> We just wanted to follow up regarding our submitted rebuttal. As the discussion period closes soon (by August 6th, AoE), we would greatly appreciate the opportunity to clarify or address any remaining questions or concerns you might have. If there is anything further we can provide, we are more than happy to do so.
> If everything looks good on your end, we would be truly grateful if you would consider raising the score. We sincerely appreciate your support.

---

### Note · Authors · 2025-08-12

We thank all reviewers for the valuable feedback and constructive discussions during the review process. We are glad that our main contributions have been acknowledged: a systematic audit of KGQA datasets, a scalable framework for high-quality KGQA benchmark construction, and the release of KGQAGen-10k to support robust and verifiable evaluation.

Following the suggestions, we have added performance comparisons across WebQSP, CWQ, and KGQAGen-10k, showing that existing benchmarks often overestimate model performance due to outdated facts and annotation issues, while KGQAGen-10k shows model limitations more clearly. We also conducted comprehensive ablation studies covering subgraph selection strategies, different LLMs, and the role of SPARQL-based answer validation. The results highlight the effectiveness of our guided exploration and symbolic verification in improving generation quality and answer precision.

To address concerns on task difficulty, we included a graph-based complexity analysis following the DyVal framework. The statistics confirm that KGQAGen-10k contains diverse and non-trivial reasoning graphs, making it suitable for future evaluations of KG-RAG models.

Regarding LASM, we clarified that it is used only when exact match fails, and we manually verified its reliability. Among 100 sampled cases, it showed a 1% error rate, supporting its use as a fallback. We also connected it with recent work on LLM-as-a-Judge in NLG and KGQA evaluation.

Lastly, we demonstrated that KGQAGen is not limited to Wikidata. We applied it to DBpedia with minimal changes and achieved high validation quality, confirming the generalizability of our framework.

We appreciate the thoughtful suggestions throughout the process (e.g., merging appendix content, updating references to published versions). We will incorporate all feedback into the final version.

Thank you again for your time and support.

---

### Decision · Program_Chairs · 2025-09-18

**Decision:**

Accept (poster)

**Comment:**

This paper identifies the problem of existing KGQA data sets, where the average factual correctness rate is only 57%, and presents a large-scale KGQA benchmark named KGQAGen by leveraging structured knowledge wikidata, LLM, and symbolic verification. A variety of KG-RAG models are also evaluated on this large-scale benchmark, showing the limitation of existing KG-RAG methods and calling for new methods for KG-based reasoning.


All reviewers appreciate the contribution of this benchmark and vote for acceptance.